# A gravity-based three-dimensional compass in the mouse brain

Dora E. Angelaki[1,2 ✉], Julia Ng[2], Amada M. Abrego[2], Henry X. Cham[2], Eftihia K. Asprodini[3], J. David Dickman[2,4] & Jean Laurens [2]

Gravity sensing provides a robust verticality signal for three-dimensional navigation. Head direction cells in the mammalian limbic system implement an allocentric neuronal compass. Here we show that head-direction cells in the rodent thalamus, retrosplenial cortex and cingulum fiber bundle are tuned to conjunctive combinations of azimuth and tilt, i.e. pitch or roll. Pitch and roll orientation tuning is anchored to gravity and independent of visual landmarks. When the head tilts, azimuth tuning is affixed to the head-horizontal plane, but also uses gravity to remain anchored to the allocentric bearings in the earth-horizontal plane. Collectively, these results demonstrate that a three-dimensional, gravity-based, neural compass is likely a ubiquitous property of mammalian species, including ground-dwelling animals.

[1] Center for Neural Science and Tandon School of Engineering, New York University, New York, NY, USA. [2] Department of Neuroscience, Baylor College of Medicine, Houston, TX, USA. [3] Department of Pharmacology, Faculty of Medicine, School of Health Sciences, University of Thessaly, Larissa, Greece. [4] Department of Electrical and Computer Engineering, Rice University, Houston, TX, USA. ✉email: da93@nyu.edu

Gravity is a ubiquitous force that profoundly affects life on earth. Gravity assists or resists movements[1,2], accelerates free-falling objects such as a ball[3] and shapes our habitations' architecture. As such, graviception represents one of the most ubiquitous sensory modalities of living organisms[4,5].

Animal species across a wide range of classes[6] orient themselves and navigate in three dimensions (3D). Preeminent neuronal classes of the mammalian's brain, such as place cells[7,8] and head direction (HD) cells[9] operate in 3D. By providing verticality information[10,11], gravity may mitigate the complexity of orienting in 3D[6,12]; yet gravity signals have never been identified in the brain's navigation system. Here we tested whether mouse HD cells, which encode allocentric head orientation analogous to a neural compass, use gravity-anchored tilt signals (orientation relative to vertical) and azimuth signals (orientation in the gravity-horizontal plane, measured in a so-called tilted frame[13–15] during 3D motion, Fig. 1a) to yield a sense of 3D head orientation.

Unlike bats[9], tuning to tilt has never been shown in rodents, and some researchers report that it may be absent in ground-dwelling species like rodents[16,17]. Thus, we first show that HD cells in the mouse anterior thalamus and retrosplenial cortex are tuned to combinations of azimuth and tilt. We also confirm that 3D HD signals travel across brain regions by recording from the cingulum fiber bundle, which connects areas of the navigation system[18]. Next, we present a conceptual and mathematical framework to model 3D HD responses, where tilt and azimuth tuning interact multiplicatively to encode 3D orientation. Finally we show that, not only does gravity anchor tilt tuning, but it also defines the earth-horizontal plane to which the azimuth compass is referenced[14]. Thus, a 3D, gravity-based orientation compass is not a specialized property limited to areal species but may instead be ubiquitous throughout many chapters of animal evolution.

## Results

**A 3D compass in the mouse brain.** We used tetrodes to record extracellularly from the antero-dorsal nucleus of the thalamus (ADN; $n = 4$ mice; Supplementary Table. 1), retrosplenial cortex (RSC; $n = 4$ mice; Supplementary Table. 1), and cingulum fiber bundle (CIN; $n = 4$ mice; Supplementary Table. 1). The CIN carries projection fibers from the ADN and RSC[18–21]. Cells were exclusively selected based on spike isolation (Supplementary Fig. 1) and recording locations were verified post-mortem (Supplementary Fig. 2). On the basis of their responses during free foraging in a horizontal arena (Fig. 1b; summarized in Supplementary Fig. 3), cells were characterized as azimuth-tuned (i.e. traditional HD) cells in light (Fig. 1c, red) and darkness (see example in Fig. 1c, black) or azimuth-untuned (see example in Fig. 1d).

Neurons were then characterized as animals walked on a platform orientable in 3D (Fig. 1e) that could be tilted up to 60° (Supplementary Fig. 4). We represented tilt tuning curves in spherical coordinates, with 2 degrees of freedom: absolute tilt angle from upright: α (range: 0–180° in the pitch, roll or intermediate planes), and tilt orientation: γ (range: 0–360°; see Supplementary Fig. 5 for definitions of right/left ear-down [RED/LED; roll plane] and nose-up/nose-down [NU/ND; pitch plane] orientations). For simplicity of illustration, 2D tilt tuning curves are shown in a planar representation using an equal-area Mollweide projection, and the 1D azimuth axis is unfolded.

Next, the neurons' tuning was plotted as a color map in a volume formed by this tilt plane and the azimuth axis (Fig. 1f, g; Supplementary Movie 1). Both example neurons in Fig. 1 exhibited tilt tuning, characterized by an increased firing rate at a preferred tilt (Fig. 1f, g, NU for both cells; see Supplementary Movie 2), with peak-to-trough amplitudes of 32 Hz and 23 Hz,

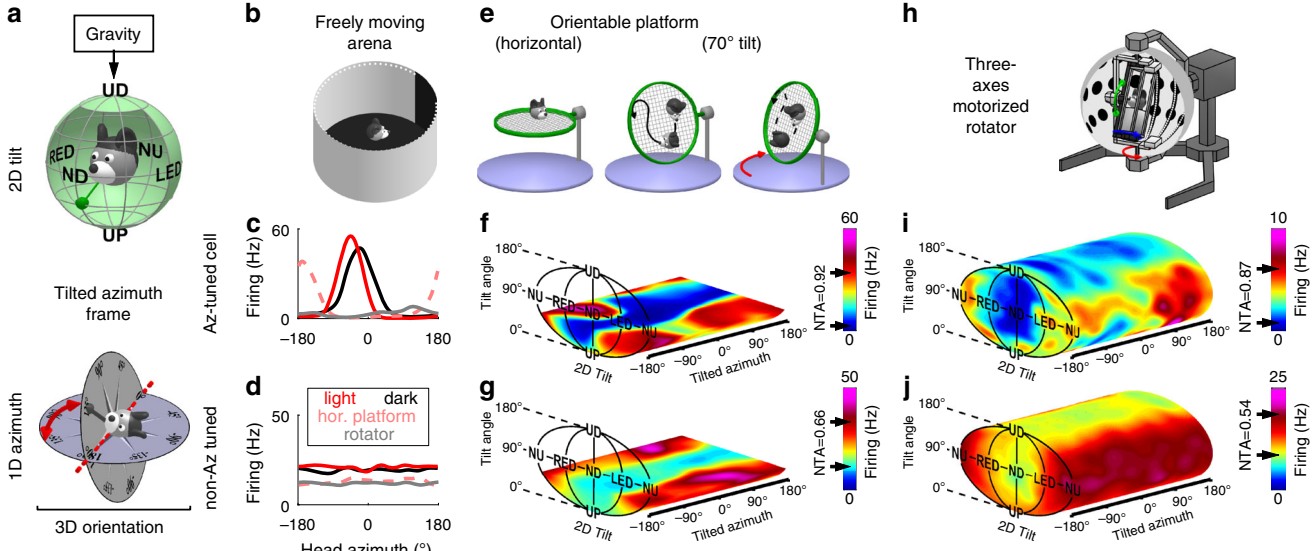

**Fig. 1 Three-dimensional response of two example cells. a** Proposed framework for 3D orientation. Top: tilt is measured by sensing the gravity vector (green pendulum) in egocentric coordinates, resulting in a 2D spherical topology. Bottom: azimuth has a circular topology, and is measured by rotating an earth-horizontal compass in alignment with the head-horizontal plane (TA frame). **b** Schematic of the arena used to identify azimuth-tuned cells in the horizontal plane. **c, d** Example azimuth tuning of a traditional HD cell, i.e. tuned to azimuth (Az-tuned) in the ADN (**c**) and another cell not tuned to azimuth (non-Az-tuned cell) in the cingulum (**d**), as the mouse walks freely in light (red) and darkness (black) in a horizontal arena (shown in **b**), on a platform oriented horizontally (shown in **e**, left; broken pink lines) and in the rotator (shown in **h**; gray lines). The azimuth-tuned cell showed significant tuning with different preferred directions (PD) in all setups, although response was strongly attenuated in the rotator (compare gray with red/pink lines). **e** Schematic of a 3D orientable platform used to measure 3D tuning. **f, g** Tuning curves for the two cells in **c** and **d**, obtained from responses as the mouse foraged on the orientable platform (shown in **e**). Firing rate is shown as a heat map in 3D space (Supplementary Movies 1, 2). The peak and trough of the average tilt response (across all azimuths) are indicated with arrows on the color scale; NTA = (peak−trough)/peak. Note that tuning curves are restricted to 60° tilt (Methods). **h** Schematic of a rotator used to measure full 3D tuning curves. **i, j** Tuning curves for the two cells in **c**, **f** and **d**, **g** as the mouse was passively re-oriented uniformly throughout the full 3D space using the rotator (Supplementary Movies 3–5).

respectively, when averaged across all azimuth angles. For better comparison between cells, we divided these amplitudes by the cells' peak firing to compute their normalized tuning amplitudes (NTA; 0.92 and 0.66, respectively). In addition, with the platform in the earth-horizontal orientation, the cell classification as azimuth-tuned or azimuth-untuned persisted (Fig. 1c, d, dashed pink curves).

Next, animals were transferred to a multi-axis rotator (Fig. 1h) that sampled neurons' full 3D tuning curves uniformly (Supplementary Fig. 6; Supplementary Movie 3). The example azimuth-tuned cell in Fig. 1c, f maintained both azimuth (Fig. 1c, gray) and tilt (Fig. 1i; see also Supplementary Movie 4) tuning, although its peak firing rate had decreased from ~60 to ~10 Hz. While the example cell's preferred direction (PD) in azimuth differed across environments (Fig. 1c, compare solid red, dashed pink and gray lines, with PDs at −46°, −163°, and 114°, respectively), its tilt PD was relatively constant at NU (Fig. 1i: [$\alpha = 83°$, $\gamma = 180°$], as compared to Fig. 1f: [$\alpha = 56°$, $\gamma = 130°$; note that tuning was not sampled at higher tilt angles]). Similarly, the example azimuth-untuned cell exhibited a tilt PD at [$\alpha = 97°$, $\gamma = 168°$] (Fig. 1j; Supplementary Movie 5), as compared to [$\alpha = 48°$, $\gamma = 150°$] (Fig. 1g). Thus, tilt tuning recorded when animals were passively re-oriented had similar spatial properties to that when moving freely.

We used identical criteria to classify neurons as azimuth-tuned or tilt-tuned (Methods; Supplementary Fig. 7). When tested on the platform, tilt tuning was widespread in azimuth-tuned (traditional HD) cells classified based on their responsiveness in the horizontal arena (of Fig. 1b), as summarized in Fig. 2. Specifically, out of 29 ADN neurons recorded on the platform, 25 (86%) were classified as azimuth-tuned (Fig. 2a, Venn diagram). Among these, 24 (96%) were also tuned to tilt and are subsequently called conjunctive (azimuth and tilt) HD cells (solid red symbols in Fig. 2). A sizeable population of azimuth-tuned cells were also recorded in RSC and CIN (49% and 40% of recorded cells, respectively). Of these, 58% (RSC) and 76% (CIN) were conjunctive cells.

Tilt tuning on the platform was also seen in azimuth-untuned cells (solid black discs and symbols in Fig. 2a). Thus, tilt tuning was common, observed in all areas, regardless of azimuth tuning, with 92/139 (66%) tilt-tuned cells. A total of 75/139 (54%) cells were azimuth-tuned, and tilt and azimuth tuning overlapped across neurons. Tilt-tuned cells were slightly (7%) more likely to be azimuth-tuned and reciprocally azimuth-tuned cells were slightly (8%) more likely to be tilt-tuned (chi-square test, $p = 0.02$, $\chi^2 = 0.02$, 1 dof). The NTA of tilt was lower than that of azimuth in conjunctive ADN cells, and similar in other regions (Fig. 2a; Wilcoxon-paired rank test; $p = 10^{-3}$ in ADN, $p = 0.2$ in RSC, $p = 0.9$ in CIN, see also Supplementary Fig. 7d–f). These results indicate that tilt signals are an inherent component of the mouse HD system during natural behavior; thus, the term HD cell should refer to both tilt-tuned cells as well as azimuth-tuned cells.

**Head-direction tuning in the full 3D space**. To characterize tuning uniformly in 3D space, 549 (60 ADN, 202 RSC, 287 CIN) neurons were tested in the rotator. Seventy-one percent (388) of these cells were significantly tuned to tilt (88% ADN, 66% RSC and 70% CIN). Similar to responses obtained on the platform, tilt-tuned cells were slightly (5%) more likely to be azimuth-tuned and reciprocally azimuth-tuned cells were slightly (7%) more likely to be tilt-tuned (chi-square test, $p = < 10^{-3}$). Before analyzing the 3D properties of HD tuning, we first verified that spatial responses were similar in freely moving animals and in the rotator.

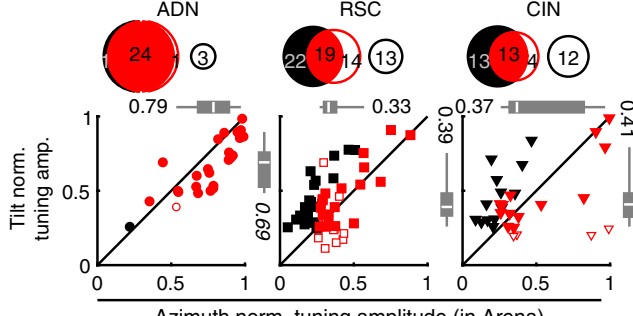

**a** Population responses in freely moving animals

ADN    RSC    CIN

Tilt norm. tuning amp.

Azimuth norm. tuning amplitude (in Arena)

Population responses in restrained animals

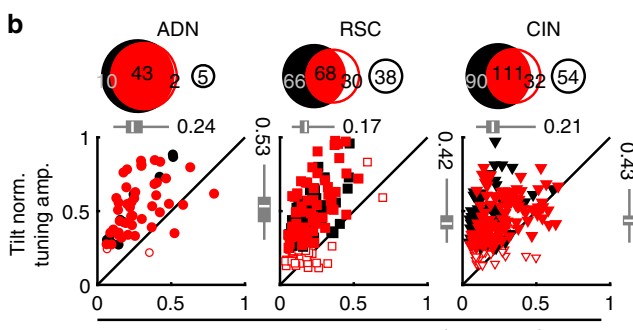

**b**   ADN    RSC    CIN

Tilt norm. tuning amp.

Azimuth norm. tuning amplitude (in Rotator)

Response properties when freely moving versus restrained

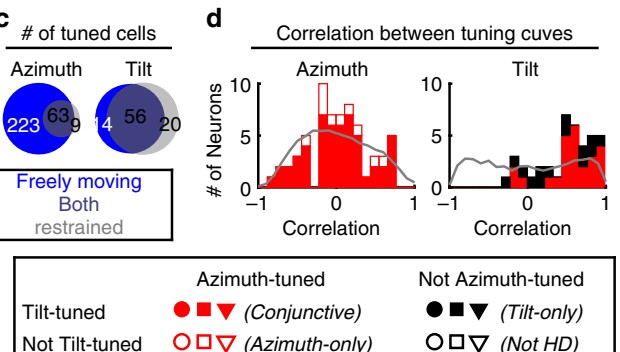

**c** # of tuned cells

Azimuth    Tilt

223  63  9     4  56  20

Freely moving
Both
restrained

**d** Correlation between tuning curves

Azimuth    Tilt

# of Neurons

Correlation    Correlation

|  | Azimuth-tuned | Not Azimuth-tuned |
|---|---|---|
| Tilt-tuned | ● ■ ▼ (Conjunctive) | ● ■ ▼ (Tilt-only) |
| Not Tilt-tuned | ○ □ ▽ (Azimuth-only) | ○ □ ▽ (Not HD) |

In line with previous studies[22,23] and the example cell in Fig. 1c, i, azimuth tuning modulation amplitude was attenuated when animals were restrained in the rotator (scatterplots in Fig. 2b vs. Fig. 2a; Supplementary Fig. 8a, b). As a result, only a minority of neurons tuned to azimuth when moving freely were significantly tuned to azimuth in the rotator (63/286; Fig. 2c, left panel). Nevertheless, the PDs of multiple azimuth-tuned HD cells had consistent angles relative to each other when moving freely and in the rotator (Supplementary Fig. 8f), indicating the structure of the population of azimuth-tuned cells was maintained in the rotator. Thus, other than the smaller magnitude, azimuth responses measured in the rotator are representative of the neurons' natural responses.

Tilt tuning magnitude was also attenuated in the rotator, although to a lesser extent (Supplementary Fig. 8c, d). A minority of cells were only tilt tuned in the rotator because of the larger sampling of 3D space. Furthermore, some cells were significantly tuned only when moving freely because the response magnitudes were larger (Supplementary Fig. 8c, d). Nevertheless, a large proportion of cells (62%) were tuned to tilt under both conditions

**Fig. 2 Population azimuth and tilt tuning in freely moving vs. restrained animals. a** Summary of tuning prevalence during unrestrained motion. Azimuth tuning was derived from data in the freely moving arena (Fig. 1b). Tilt tuning was derived from data on the 3D platform (Fig. 1e). For each panel, Venn diagrams (top) indicate the number of tilt-tuned (filled black discs) and azimuth-tuned (red discs) cells. Conjunctive cells appear at the intersection of these discs. Open discs illustrate cells responsive to neither tilt nor azimuth. The scatterplots (bottom) indicate the normalized modulation amplitude of responsive cells. The boxes and whiskers represent the median (white line), 95% confidence interval (boxes) and upper/lower quartiles (whiskers) of the azimuth modulation of azimuth-tuned cells (top) and tilt modulation of tilt-tuned cells (right). Different symbols (based on recorded area) are color-coded based on cell type (Conjunctive: filled red; Azimuth-only: open red; Tilt-only: filled black). **b** Prevalence of tilt tuning in the rotator (Fig. 1h) and azimuth tuning (when moving freely). Format as in **a**. **c** Comparison of responsiveness for cells tested in both restrained and freely foraging animals. Venn diagrams with the number of cells tuned when moving freely (blue) in the arena (azimuth tuning) or 3D platform (tilt tuning) and restrained in the rotator (gray). Cells tuned under both conditions appear at the intersection of both discs. **d** Pixel-by-pixel correlation of the fitted azimuth (left) and tilt (right) tuning curves (only cells tuned under both freely moving and restrained conditions are included). For tilt tuning, the rotator data were re-analyzed by restricting tilt angles up to 60° (to match the conditions in the platform). Gray: expected distribution if tuning curves shift randomly (H0), computed by randomly shuffling the cells. Source data are provided as a Source Data file.

(Fig. 2c, right panel), indicating that tilt tuning is conserved across free locomotion and restrained, passive motion conditions.

We then compared tuning curves when moving freely and in the rotator by computing their pixel-by-pixel correlations. This revealed an important difference between azimuth and tilt tuning. Because azimuth curves shifted randomly between environments (Supplementary Fig. 8e), their pixel-by-pixel correlations were uniformly distributed (median = 0.07; Kolmogorov–Smirnov test $p = 0.15$; Fig. 2d, left). In contrast, tilt tuning was preserved (median = 0.58; $p < 10^{-5}$; Fig. 2d, right), as expected if the tilt compass was anchored to a common reference: gravity (Fig. 1a; see below).

**Azimuth tuning in 3D**. To investigate how tilt and azimuth components work together to encode 3D head orientation, we first questioned how to define azimuth when the head tilts away from upright. The brain may simply project head direction onto the earth-horizontal (EH) plane and encode azimuth in that plane (Fig. 3a). Alternatively, it may measure the orientation the head would have if it were rotated back to upright (Figs. 1, 3b, Supplementary Fig. 9a), which is equivalent to rotating the EH compass to align with the head-horizontal plane, resulting in a tilted azimuth (TA) compass[13,14]. Early models[16,17] proposed that azimuth is updated in the head-horizontal plane by tracking rotations in this plane (yaw; Fig. 3c, cyan), ignoring other movements (Yaw-only model, YO). However, this would not maintain allocentric invariance in 3D[14]. For example, when completing the trajectory in Fig. 3c (red), the compass would register only three right-angle turns (Fig. 3c, cyan), i.e. 270°. To maintain allocentric invariance, a TA compass must use a dual updating rule[13], which includes both yaw (Fig. 3c, cyan) and earth-horizontal rotations (Fig. 3c, green). In the example of Fig. 3c, this allows totaling 360° when completing the trajectory. Thus, we emphasize that a YO compass would loose allocentric invariance during 3D motion, even when returning to upright

(see for example Fig. 3c). In contrast, EH and TA frames remain invariant when the head tilts[14] (Supplementary Movie 6).

The 3D motion protocol allows testing the YO, EH and TA models. First, we expressed azimuth in all three frames and tested whether cells were significantly tuned when the 3D trajectory brought the head close to upright (<45° tilt). As predicted, almost no cells exhibited significant tuning in a YO frame (6/285 tuned cells, consistent with false positives at $p = 0.01$). In contrast, 63/285 (22%) cells were tuned when azimuth was expressed in either the EH or TA frames (ADN: $n = 17$; RSC: $n = 7$; CIN: $n = 39$; this relatively low percentage of significantly tuned cells is due to the attenuation of azimuth responses in the rotator, see Supplementary Fig. 8a, b).

Second, when expressed in the appropriate reference frame, the cells' azimuth PD should be invariant at all head tilts. To test this, we compared the cells' azimuth tuning curves near upright (<45° tilt) or when tilted (>60° tilt) (Fig. 3d). We observed that these curves were highly correlated when expressed in a TA frame (median: 0.73; [0.63–0.86] CI), but significantly less so when expressed in a EH frame (0.19; [0.04–0.37] CI). In addition, correlations were near zero when expressed in a YO frame (0.08, [−0.07–0.18] CI). We verified (three-way ANOVA, difference between reference frames: $p < 10^{-10}$, $F_{2,183} = 33.3$) that correlations were similar across recorded areas ($p = 0.6$, $F_{2,183} = 0.46$) and tilt-tuned or non-tilt-tuned cells ($p = 0.2$, $F_{1,183} = 1.71$). As expected (Supplementary Fig. 9a, c), expressing azimuth in a EH frame leads to a reversal of the cells' PD when pitching beyond 90° (Supplementary Fig. 9b, d, similar to previous observations[9]), but not when rolling (Supplementary Fig. 9b, d). In contrast, azimuth PDs are invariant in a TA frame and therefore this reversal did not occur (Supplementary Fig. 9b, d).

We also found that, regardless of reference frame, azimuth tuning decreased when the animal was tilted beyond 90° from upright (Supplementary Movies 6–9). As illustrated with an example azimuth-only cell in Fig. 3e (animated curve in Supplementary Movie 7), azimuth tuning was strong (PD at −85°) for small tilt angles (lowest portion of the tuning curve) but vanished at large tilt angles (i.e. upper portion of the 3D tuning curve, Fig. 3e). This was consistent for all azimuth-only cells: the average HD tuning curve (computed in a TA frame and aligned to peak at PD = 0°) had a higher modulation when computed for head tilts close to upright (Fig. 3f, red) and almost no modulation close to upside-down (Fig. 3f, gray). Thus, for azimuth-only cells, the response amplitude of azimuth tuning was dependent on the tilt angle, even though the cells were not tuned to tilt.

This was not limited to azimuth-only cells (Fig. 3g, gray): the azimuth tuning amplitude of conjunctive cells decreased similarly, irrespective of whether the cell's tilt tuning favored upright orientation (Fig. 3g, red), intermediate tilt (i.e. 90°; Fig. 3g, blue) or upside-down (Fig. 3g, green). Normalized tuning amplitudes for all azimuth-tuned cells were affected by tilt angle (two-way ANOVA, $p < 10^{-10}$, $F_{12,767} = 61$) and varied between groups of cells ($p < 10^{-10}$, $F_{3,767} = 12.8$); however there was no significant interaction effect ($p = 0.9$, $F_{36,767} = 0.42$), indicating that the azimuth tuning of all cells was equally affected by tilt. We conclude that HD cells encode azimuth in a TA reference frame, and that their azimuth response decreases when the head tilts away from upright, irrespective of tilt tuning.

**Tilt and azimuth tuning follow multiplicative interaction**. To further understand 3D tuning, we created a 3D HD model that incorporates the following properties (Fig. 4): (1) tilt tuning curves are generated by feeding the gravity vector (or any other reference vertical vector) into Gaussian tuning functions (Fig. 4a), (2) azimuth-tuned cells encode TA with a tilt-dependent gain,

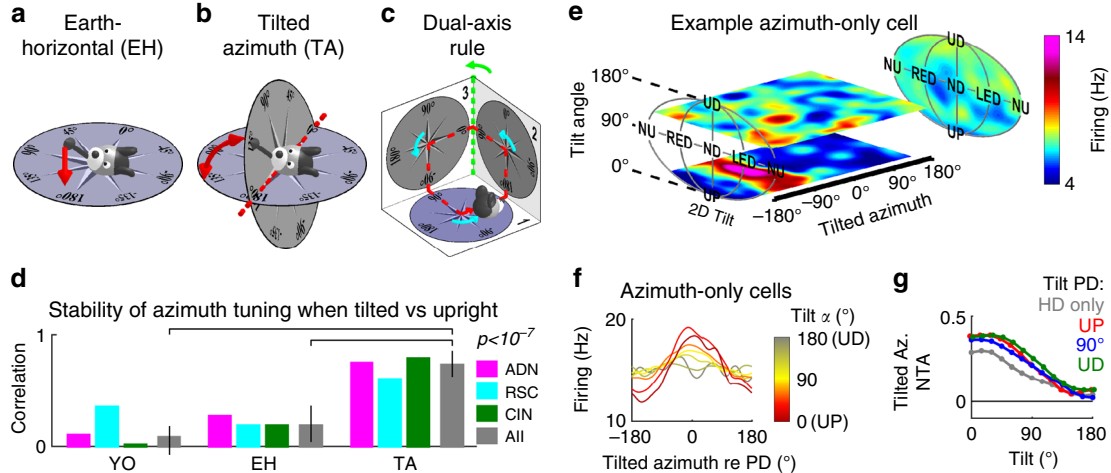

**Fig. 3 Encoding of azimuth is also influenced by gravity. a** Illustration of the earth-horizontal (EH) frame, where azimuth direction is projected onto the earth-horizontal plane (dashed red line). **b** Tilted azimuth (TA) frame[14], where azimuth is defined by rotating head direction (gray vector) towards the horizontal plane instead of projecting it. The head has the same 3D orientation in **a** and **b** but its azimuth is different in the two frames (EH: 165°; TA: 135°). **c** Dual-axis rule for updating TA, illustrated by an example trajectory (red) where the animal travels in 3D across three orthogonal surfaces (numbered 1 to 3). Head azimuth is updated when the head rotates in yaw within one surface (first rule, cyan arrows) or in the earth-horizontal plane (second rule, green arrow when transitioning from surface 2 to 3). The first rule tracks azimuth and the second rule ensures that the brain compass always matches the EH compass along the line intersecting the two planes (the 0–180° line in **b**). **d** Correlation between azimuth tuning curves when upright (<45° tilt) vs. tilted (>60°) in YO, EH and TA frames. Colored bars: Median correlation in each brain region. Lines: 95% confidence intervals. Gray bars: data pooled across all cells. p-value based on a Wilcoxon-signed rank test across all cells. **e** 3D tuning curve of an example azimuth-only cell, showing horizontal slices at tilt angles of 30° and 110° (see animations in Supplementary Movie 7). The average tilt tuning (across all azimuths) is shown on the right panel. **f** Average azimuth tuning curve of all azimuth-only cells that maintained their azimuth tuning in the rotator (n = 10), averaged across all tilt orientations (γ), computed at tilt angles (α) ranging from 0 to 180° and centered on PD = 0°. **g** Average normalized tuning amplitude of the azimuth tuning curve as a function of tilt angle, computed for azimuth-only cells (n = 10, gray) and conjunctive cells with preferred tilt directions near upright (<75° tilt, red, n = 16), near 90° tilt (±15°, blue, n = 22) and near upside-down (>105°, green, n = 15). Source data are provided as a Source Data file.

independently from tilt tuning (Fig. 4b), (3) tilt and azimuth tuning interact multiplicatively (Fig. 4c; an additive model performed worse; Methods).

In general, gravity is a 3D vector, sensed in egocentric 3D Cartesian coordinates (e.g. by the vestibular system; Fig. 4a, top), but can be restricted to a sphere surrounding the head (Fig. 4b) because its magnitude on earth is constant. In the proposed model, we applied a Gaussian tuning in 3D Cartesian coordinates before restricting the tuning curve to a spherical space. Remarkably, this allowed modeling the tuning curve of 36% of the recorded cells that peaked at two distinct head orientations, for instance NU and ND (Supplementary Fig. 10). Thus, the proposed model of Fig. 4a reflected the sensory processes underlying gravity sensation while parsimoniously accounting for seemingly complex tilt tuning. Note though that the model does not necessarily assume that tilt tuning is anchored to gravity, as another reference vertical could be used as an input.

This model could fit conjunctive cells well (Fig. 4d; n = 16 ADN, 4 RSC, 33 CIN; median ρ = 0.88, [0.86–0.91] CI), as illustrated with two examples (Fig. 4e, f; Supplementary Movies 8, 9). The peak tilt response of the first example cell occurred at a tilt angle α = 42°, at which azimuth tuning (PD = −27°) had not yet attenuated. Consequently, the cell exhibited tilt and azimuth tuning simultaneously at this tilt angle, resulting in a preferred 3D orientation, visible on both the measured (Fig. 4e, top) and fitted (Fig. 4e, bottom) curves. This was characteristic of conjunctive cells with PDs close to upright.

The second example cell exhibited a PD at a large tilt angle (α = 105°), where azimuth tuning had already substantially decreased. As a consequence, the cell appeared azimuth-tuned at small tilt angles, where tilt tuning was minimal (Fig. 4f, lower horizontal plane) and tilt-tuned at large tilt angles (Fig. 4f, upper

horizontal plane). This was characteristic of conjunctive cells with a large preferred tilt angle.

Model fits were significantly lower when 3D curves were computed with azimuth in an EH frame (Supplementary Fig. 11), confirming that the TA frame captures the cell's response better than either the EH or YO frames could. The same model also fitted 3D tuning curves for azimuth-only (median ρ = 0.75, [0.7–0.8] CI) and tilt-only cells (median ρ = 0.87, [0.85–0.88] CI).

**Spatial properties of tilt tuning.** As illustrated in Fig. 5a, the PDs of tilt-tuned cells were widely scattered. Yet, the distribution was not uniform, with an over-representation of PDs around ND and an underrepresentation of PDs in the roll (LED/RED, gray sectors) plane (chi-square test, p < 10⁻⁵ in AND, p < 10⁻⁷ in RSC, p < 10⁻⁹ in CIN; χ² = 28; 37; 45, respectively; 3 dof). In contrast, PDs were distributed uniformly between tilt angles lower or higher than 90° (chi-square test, p = 0.4 in AND, p = 0.016 in RSC, p = 0.1 in CIN; χ² = 0.47; 5.8; 2.6, respectively; 1 dof). The gravity tuning curve of cells with PD located in the pitch (NU/ND, white sectors) plane had stronger peak firing rate (Fig. 5b; median = 12.4 vs. 7 Hz) than those with PD in the roll plane. However, both cell types have similar tuning amplitude relative to their peak firing rate (i.e. NTA; Fig. 5c). These results, showing a dominance of pitch-tuned over roll-tuned cells, are consistent with those previously described for bats[9] and monkeys[24].

Azimuth tuning of HD cells persists in darkness[25,26]. To test whether tilt tuning also persists, we recorded the responses of 210 (23 ADN; 54 RSC; 133 CIN) tilt-tuned cells in complete darkness. Angular differences between PDs recorded in light and darkness were close to zero (PD difference <45° in 151/210 cells, Kolmogorov–Smirnov test vs. the expected distribution if PD were uniformly distributed: p = 10⁻¹¹ in all areas, Fig. 5d). In addition, the tilt modulation amplitude was highly correlated

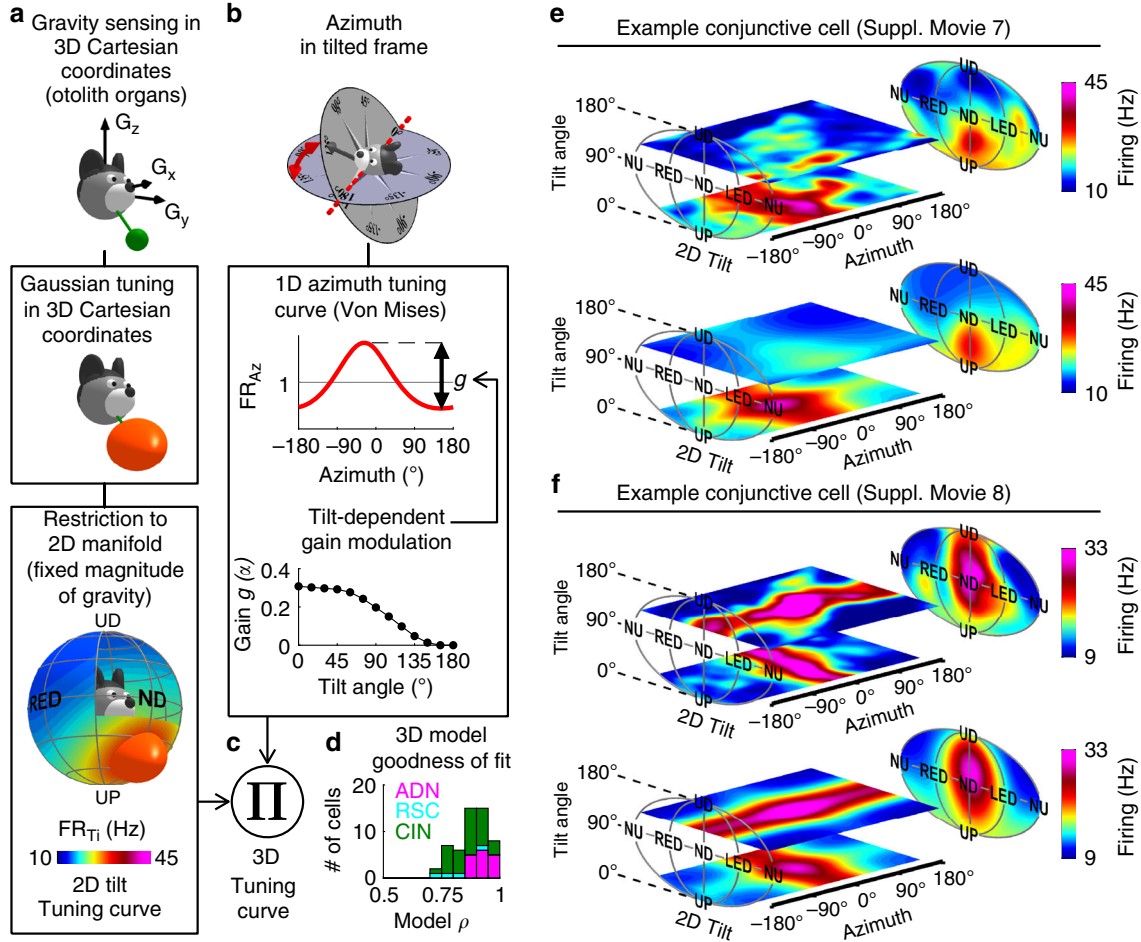

**Fig. 4 Modeling 3D responses. a** Modeling tilt tuning. Top: Gravity is a 3D vector (green) sensed in egocentric Cartesian coordinates by the otolith organs. Middle: to model tilt tuning, we first assume a 3D Gaussian function (orange ellipsoid) in this Cartesian space. Bottom: on earth, the magnitude of gravity is constant. Therefore, we restrict the tilt tuning curve to a 2D sphere surrounding the head, which corresponds to the egocentric gravity vector experienced when tilting on earth. **b** Modeling azimuth tuning. Azimuth is expressed in a TA frame (top), and tuning is modeled as a von Mises distribution combined with a tilt-dependent gain factor (Fig. 3g). **c** 3D tuning defined by the product of these two curves. **d** Distribution of the model's coefficient of correlation ($\rho$) across areas. **e**, **f** Experimentally measured 3D tuning curves (top) from two conjunctive cells that maintain their azimuth tuning in the rotator and fitted tuning curves (bottom), represented as color maps in 3D space (animated in Supplementary Movies 8, 9). The cell is **e** is the same as in **a**, **b**. Source data are provided as a Source Data file.

between light and dark conditions (Fig. 5e, Spearman correlation $r$: ADN: $r = 0.8$, slope $= 0.9$; $p < 10^{-5}$; RSC: $r = 0.88$, slope $= 0.97$; $p < 10^{-7}$; CIN: $r = 0.71$, slope $= 0.9$; $p < 10^{-10}$; the slopes are similar in cells tuned in the pitch and roll plane: 0.91 vs. 0.92), although slightly lower in darkness in CIN (paired Wilcoxon test: $p < 10^{-4}$; $p > 0.1$ in other areas). Similar findings were also reported in gravity-tuned cells in the monkey anterior thalamus[24], suggesting that these tilt-tuned neurons may be found along a broad range of animal evolution.

In further agreement with findings in monkeys[24], a small fraction of cells also responded to tilt or azimuth velocity (Supplementary Fig. 12). In addition, tilt tuning in the rotator could be reproduced using traditional single-axis rotations like pitch and roll (Supplementary Fig. 13). Thus, tilt tuning is anchored to allocentric space, independent of the exact motion trajectory. Finally, we verified that 3D tilt tuning curves were highly reproducible across repetitions of the rotation protocol across different days (Supplementary Fig. 14).

**3D tuning is anchored to gravity**. The invariance of tilt tuning in light and dark conditions (Fig. 5d, e), and across setups (Fig. 2d), supports the hypothesis that gravity—rather than visual landmark

cues—represent the allocentric vertical reference for tilt tuning. Yet, the model in Fig. 4a does not strictly assume that tilt signals originate from gravity—mathematically it could apply to any vertical reference, such as the visual scene, or a combination of gravitational and visual cues.

A distinct but related question is whether gravity anchors TA tuning. During 3D motion, azimuth is measured in a TA compass defined by rotating the earth-horizontal compass in alignment with the head-horizontal plane. But is that earth-horizontal plane defined by visual or gravitational cues?

To test these hypotheses, we recorded 148 (22 ADN; 46 RSC; 80 CIN) tilt-tuned cells with the 3D rotation protocol after tilting the rotator and visual surround together 60° (Fig. 6a, protocol 3T). This dissociated the vertical axis defined by the visual cues inside the sphere (Fig. 6a, blue) from the gravitational vertical (Fig. 6a, green; see Supplementary Fig. 15a–e). We first investigated which modality anchors tilt tuning, and tested which modality anchors the TA signal in a second step.

To answer the first question, we assumed that tilt-tuned HD cells are referenced to a weighted mean (weight $w$) of gravity and vision (Fig. 6a, black). At this stage, we assume that the TA signal is anchored to gravity. We computed each cell's tuning curve for

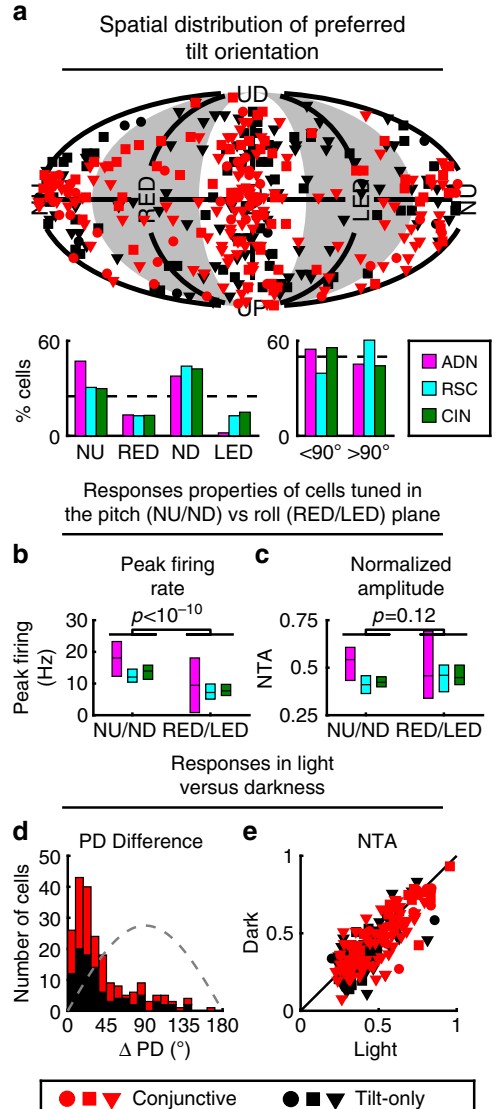

**Fig. 5 Summary of 3D tilt tuning (based on full tuning curves measured in the rotator). a** Top: Distribution of tilt PDs. Red: Conjunctive (azimuth and tilt) cells; Black: tilt-only cells. Circles, squares, triangles: ADN, RSC, CIN, respectively. Bottom: Number of cells with PD in the roll (RED/LED) or pitch (ND/NU) plane; and in upper (<90° tilt) or lower (>90° tilt) hemisphere, color-coded separately for each area (AND, CIN, RSC). **b**, **c** Comparison of peak firing rate (**b**) and NTA (**c**) of the tilt tuning curves of neurons with PD in the roll and pitch plane, color-coded by regions. Boxes represent the median and 95% confidence interval of the median. p-values are based on a Wilcoxon-rank sum test. **d** Distributions of absolute difference in tilt PD for light vs. darkness for conjunctive (red) and tilt-only (black) cells. Gray dashed line: expected distribution if PDs are independently distributed on a sphere. **e** Comparison of tilt peak-to-trough normalized tuning amplitude computed from Gaussian fits (Methods) in darkness vs. light. Red: Conjunctive (azimuth and tilt) cells; Black: tilt-only cells. Source data are provided as a Source Data file.

each value $w$ (e.g. with $w = 0$ and $w = 1$ in Fig. 6b) and tested how it correlated with the fitted tuning curve recorded with the rotator upright (Fig. 6c), which was used as a reference since the gravity- and visually referenced verticals are identical. For the example cell in Fig. 6a–c, the correlation peaked at a value $\rho_{peak} = 0.81$ for a gravity weight $w_{peak} = 1$ (Fig. 6d, red), indicating that this cell encodes gravity-referenced tilt. At the population level, the peak gravity weight $w_{peak}$ clustered around a

median value of 1.01 (Fig. 6e, [0.95–1.04] CI; data from 125/148 cells where the peak correlation was significantly higher than 0; see Supplementary Fig. 15k for details). The peak gravity weight was identical in all recorded areas (Krusall–Wallis nonparametric ANOVA, $p = 0.17$) and between cells with PD in the pitch or roll plane (Wilcoxon-rank sum test, $p = 0.66$). We also tested whether the peak gravity weight was significantly different from 1 on a cell-by-cell basis (Supplementary Fig. 15l) and found that this was the case in only one cell (likely a false positive). Likewise, no neuron correlated better with a visually referenced frame compared to a gravity-reference frame (Supplementary Fig. 15m). Thus, our data indicates that tilt-tuned cells encode exclusively gravity-anchored tilt signals, as opposed to visually anchored signals or a mixture thereof. These findings are identical to tilt-tuned cells in the macaque anterior thalamus[24].

Next, we investigated whether TA is anchored to the earth-horizontal plane defined by visual cues. First, we repeated in analysis in Fig. 6d, e but assumed that TA is referenced to vision to confirm that the conclusion that tilt tuning is anchored to gravity still held. The correlation still peaked at a value close to 1 in the example cell (Fig. 6d, broken gray line, $w_{peak} = 0.89$) and, at the population level, $w_{peak}$ was still centered on 1.03 ([0.97–1.09] CI; Fig. 6f). Accordingly, we fixed the gravity weight $w$ to 1 in the following analysis. We computed 3D tuning curves assuming that TA is referenced to the gravity-based or visually based horizontal plane, and compared them to the curves measured with the rotator upright by computing the partial correlation of azimuth tuning (where the correlation attributable to gravity tuning is eliminated, see Methods). We analyzed 19 cells (5 ADN, 1 RSC, 13 CIN) that were tuned to azimuth when moving freely and during Experiment 3-L (same inclusion criterion than in Fig. 3) and were recorded with the rotator tilted (Fig. 6g). At the population level, correlations were higher in a gravity-referenced frame (Wilcoxon-signed rank test, $p = 5.10^{-4}$). On a cell-by-cell basis, the partial correlation was higher when TA was referenced to gravity 8 cells (3 ADN, 5 CIN) and was not significantly different between the two frames was non-significant in all other cells (markers with gray border; note that the correlations in Fig. 6g are not significantly different from 0 in 8/19 cells because azimuth tuning is weak in the rotator). We conclude that the earth-horizontal plane that anchors TA is defined by gravity, which thus provides a vertical reference for all aspects of 3D orientation.

## Discussion

In summary, these findings demonstrate that HD cells in two areas of the mouse navigation system, as well as their output fiber bundle, are tuned in 3D. HD cells encode 2D tilt either in isolation or conjunctively with 1D azimuth (Fig. 2). The spatial properties of azimuth tuning are independent of tilt tuning (Fig. 3g), and the two are separable; i.e. a cell's entire 3D head orientation tuning curve can be computed given its tilt and azimuth tuning (Fig. 4). Tilt tuning is referenced to the gravitational vertical (Fig. 6d–f). Finally, azimuth tuning is anchored to visual landmarks[25] but, during 3D motion, it is defined by rotating the gravitationally defined earth-horizontal compass in alignment with the head-horizontal compass[13,14] (Fig. 3; Fig. 6g).

A recent study by Shinder and Taube[17] concluded that HD cells encode only azimuth computed by integrating rotations in the head-horizontal (yaw) plane. However, in our assessment[15], this study is in fact supportive of the tilted azimuth model, which was not directly tested in that work. Furthermore, we argue that their study is inconclusive with respect to tilt tuning[15] (Supplementary Fig. 16).

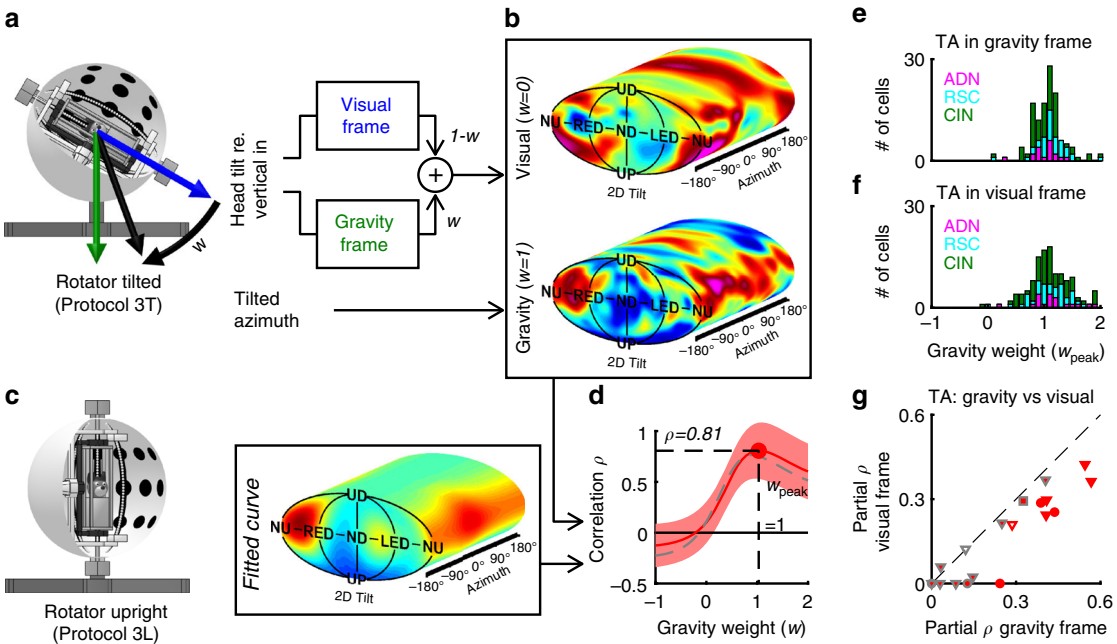

**Fig. 6 Tilt tuning is anchored to gravity. a, b** Rationale of the analysis and 3D tuning curve of an example cell when tilt is expressed relative to visually referenced vertical (**b**, upper tuning curve) or gravitational vertical (**b**, lower tuning curve). **c** Reference curve computed with the rotator upright. **d** Red: Coefficient of correlation between the tuning curves in **a** and the reference in **b**, as a function of the weight $w$, assuming TA is referenced to gravity. The red band indicates 95% confidence interval. Broken gray line: same correlation, computed assuming that TA is anchored to vision. **e, f** Gravity weight at which the correlation is maximal for all tilt-tuned cells where $w_{peak}$ is significantly higher than 0, computed when TA is referenced to gravity (**e**) or vision (**f**). **g** Comparison of the partial correlation (with the effect of gravity removed) of azimuth-tuned cells, assuming that TA is anchored to a gravity or visually referenced frame. Open/filled symbols: azimuth-only and conjunctive cells. Red/gray: cells where the difference between the two frames is significant or not. Source data are provided as a Source Data file.

The proposed 3D model is compatible with the toroid topology proposed for HD cells in bats[9] when azimuth is expressed in a TA frame and tilt is restricted to the pitch plane (Supplementary Fig. 17). Tilt PD are not uniformly representing 3D space, as the pitch plane is over-represented, consistent with previous findings in both macaques[24] and bats[9]. This over-representation is observed in tilt-only and well as in conjunctive cells, indicating that it is not linked to the 3D properties of azimuth tuning.

We found that azimuth tuning subsides when mice are close to upside-down, in agreement with previous findings in rats[16,17], possibly because upside-down in a singularity point at which TA is mathematically undefined[13] (Supplementary Fig. 9). In contrast, half of the HD cells in bats retain an azimuth tuning when upside-down[9] that may be explained by the toroidal model, where the singularity is lifted by restricting head tilt to the pitch plane. The HD system of bats may have adapted by using a simpler coordinate system to encode azimuth in upside-down orientation based on ethological demands.

A recent imaging study[27] indicates that the human RSC encodes pitch orientation in a virtual navigation task, although the ADN was found to encode mainly azimuth. It is possible that visually driven tilt signals arise in the RSC in a virtual environment where visual, but not inertial gravity cues, are present.

We conclude that 3D tuning may be a ubiquitous feature of the mammalian HD system. We suggest that the denomination head direction cell should also apply to tilt-tuned and conjunctive cells as well as previously described azimuth-tuned cells.

Together with Finkelstein et al.[9], the present study reveals that HD cells tuned in 3D exist in the ADN, RSC and presubiculum. ADN HD cells project to layer III pyramidal neurons in the presubiculum[28,29], and both populations discharge coherently[26], suggesting that presubicular HD cells may inherit their 3D properties form the ADN. The function of presubicular

HD cells likely extend beyond relaying ADN HD signals, as indicated by the presence of egocentric information in presubicular but not ADN HD cells[30] and the importance of the presubiculum for visually anchoring the HD network[31]. The RSC is involved in visual processing, often hypothesized to transform visual landmarks from an egocentric to an allocentric reference frame[32,33], and RSC HD cells may combine HD signals with visual[34] or egocentric spatial information[35,36]. Our findings (and Kim and Maguire's study[27]) raise the possibility that the RSC may use gravity-referenced tilt signals to transform visual signals in 3D.

It is notable that Finkelstein et al.[9] observed a functional gradient of azimuth-tuned to tilt-tuned HD cells in the presubiculum. We observed no difference between the granular and dysgranular RSC, nor any obvious functional gradient in the ADN (Supplementary Fig. 18).

Our study is the first to record HD cells directly in the cingulum fiber bundle that conveys ADN and RSC projections to parahippocampal regions (ADN and RSC projections), to RSC (ADN projections) and to the cingulate cortex (RSC projections)[18–21]. Recordings of axonal spikes with tetrodes are uncommon but possible[37]. Furthermore, histology clearly demonstrates that recordings occurred in white matter (Supplementary Fig. 2g–j) and units recorded in the cingulum exhibited short spike duration consistent with axonal spikes[38] (Supplementary Fig. 18b). The existence of tilt-tuned HD cells in the cingulum bundle indicates 3D signals are communicated between various regions of the limbic system.

Gravity is a fundamental vertical allocentric cue[12], which dominates vision in human verticality perception[10], even though visual signals can replace gravity cues in microgravity[39]. Further, gravity sensing represents one of the most ubiquitous sensory modalities of terrestrial living organisms[4,5]. Gravity is likely sensed

by a combination of proprioceptive and vestibular inputs[40,41], and its computation likely involves the vestibulocerebellum[42–44].

Prominent views of azimuth-tuned HD cells posit that they form a neuronal attractor that can memorize azimuth in the absence of sensory inputs[14,26,45], although some HD cells in the RSC[34], and parahippocampal regions[46] may not contribute to the attractor network. Fundamentally, the 1D attractor model implies that cells with similar azimuth PD are constrained to fire together. However, most azimuth-tuned HD cells are also tilt-tuned, and cells with similar azimuth PD may have different tilt PD (Supplementary Fig. 19c). Such cells must fire at different head position when the head tilts, contradicting the principle of an attractor. Indeed, we found that 1D attractor activity weakened when animals walked on the platform with 60° tilt (Supplementary Fig. 19), and we hypothesize that it would weaken further if tilt was increased. This suggests that the HD system follows 1D attractor dynamics when the head is upright, but this may not generalize to 3D motion. This raises the question of dimensionality of the HD system during 3D motion. Previous studies[47,48] have used unsupervised approaches to reveal the 1D attractor. We did not attempt to generalize these approaches to 3D because our data is currently restricted to 60° tilt in freely moving animals, and because HD responses are largely attenuated in the rotator.

Establishing whether tilt-tuned cells also form a 2D attractor will be challenging, especially since gravitational input that anchors tilt tuning is not easily altered. Alternatively, there may not be a gravity attractor, as there is no need to: the vestibular system can directly compute gravity orientation, and no mathematical integration may be necessary[44,49].

Tilt and 3D orientation tuning had previously only been identified in aerial (bats) and tree-dwelling (macaques) species, raising the question of whether a 3D compass would be ethologically relevant to rodents. Although laboratory mice (*Mus musculus*) and rats (*Rattus norvegicus*) are primarily land-dwelling, they exhibit a rich 3D behavioral repertoire in the wild[50–52], easily learn 3D spatial orientation tasks[53] and are physiologically related to tree-dwelling rodents[54], including other muroids (e.g. harvest mice, *Micromys minutus*) and non-muroids (e.g. squirrels). It is therefore not surprising that rodents, like bats[9] and likely macaques[24] and humans[27], possess a three-dimensional compass, whose properties may be shared across mammals.

## Methods

**Animals**. A total of 13 male adult mice (C57BL/6J), 3–6 months old, were used in this study (Supplementary Table 1). Animals were prepared for chronic recordings by implanting a head-restraint bar and a microdrive/tetrode assembly under general anesthesia (Isoflurane) and stereotaxic guidance. Two skull screws were implanted in the vicinity of the target region, and a circular craniotomy (~1.5 mm diameter) was performed above the target region. Animals were single-housed on a reversed [12/12] light/dark cycle. Experimental procedures were conducted in accordance with US National Institutes of Health guidelines and approved by the Animal Studies and Use Committee at Baylor College of Medicine.

**Neuronal recordings**. Neurons were recorded using 6 (mice AA1/AA2) or 4 (all other mice) tetrode bundles constructed with platinum-iridium wires (17 micrometers diameter, polyimide-insulated, California Fine Wire Co, USA) and platinum-plated for a target impedance of 200 kΩ using a Nano-Z (Neuralynx, Inc) electrode plater. Tetrodes were cemented to a guide tube (26-gauge stainless steel) and connected to a linear EIB (Neuralynx EIB/36/PTB). The tetrode and guide tube were attached to the shuttle of a screw microdrive (Axona Ltd, St Albans, UK) allowing a travel length of ~5 mm into the brain.

The stereotaxic coordinates for each tetrode implant was based upon Bregma as a reference point. The coordinates used to target both the ADN and the CIN were 0.2 mm posterior and 0.7 mm lateral to Bregma. The granular/dysgranular RSC were targeted by implanting 2.0 mm posterior and 0.07/0.7 mm lateral to Bregma, respectively.

Raw neuronal data was manually clustered based on spike waveform and amplitude, using custom Matlab scripts. Spike clusters with similar spike waveform and firing characteristics (inter-spike interval distribution and mean firing rate)

were attributed to a single neuron. When we observed that neurons recorded over successive days, and on a single tetrode, had similar firing characteristics and similar tuning to 3D head direction, we merged the data to avoid introducing duplicate data points in our analyses.

At the end of tetrode recordings from each animal, the brain was removed for histological verification of electrode location. The animals underwent transcardial perfusion with 4% paraformaldehyde (PFA). The brains were postfixed in 4% PFA and then transferred to 30% sucrose overnight. Brain sections (40 μm) were stained (Nissl or neutral red staining), and examined using bright-field microscopy to localize tetrode tracks (Supplementary Fig. 2). Photographs of histological slides were corrected for brightness, contrast, gamma and color balance.

**Experimental apparatus**. In order to identify traditional HD cells, we first recorded as mice explored freely in a circular arena (50 cm diameter, 30 cm height; Fig. 1b). The walls of the arena were white with a 45° black card to provide a visual orientation cue. To record tilt tuning in freely moving mice, the arena was replaced by a movable platform that was constructed by mounting an oblong nylon mesh (20 × 30 cm, 1.5 cm mesh) onto a manually operated three-axis gimbal system (Supplementary Fig. 4a). The system was placed at the center of a large cylinder (130 cm diameter, 2 m height), its door was left open during recording to provide a visual landmark and to allow the experimenter to monitor each mouse. In both systems, neuronal data were acquired at 22 kHz using a MAP system (Plexon Inc.). The microdrive's EIB was plugged to a tethered head stage that included two LEDs (one red and one infrared, 4 cm apart) for optical tracking (Cineplex, Plexon Inc.). In addition, mice's head were equipped with a digital 6-degree-of-feedom inertial measurement unit (IMU; SparkFun SEN-10121) for measuring head tilt relative to gravity. Perspective effects that could affect optical tracking when the head tilted away from horizontal were corrected based on the IMU data.

To measure the 3D orientation tuning using a uniform representation of tilt angles, we tested animals using a motorized rotator. It also allowed us to separate visual from gravity representations. We gently restrained each mouse's body and fixed its head rigidly, and placed it in the center of a rotation simulator (Supplementary Fig. 6a) composed of a motorized three-axis motion system (Axes I-III in Supplementary Fig. 6a) inside a visual surround sphere (1.8 m diameter) (Acutronics Inc., Switzerland). The inside of the sphere was painted in white, with three horizontal lines of dots (10° diameter, 30° spacing) to provide horizon and optokinetic cues. Three vertical LED stripes, affixed to black vertical bands, were placed 22.5° apart to provide a horizontal orientation cue. A fourth rotation axis (Axis IV) allowed tilting the rotator and the sphere together (sphere door closed). Neuronal data were acquired at 30 kHz using a neural data acquisition system (SpikeGadget, San Francisco, California). The position of the rotator's axes (and therefore the 3D orientation of the head) was measured with potentiometers installed in each rotation axis and digitized at 833 Hz.

All recorded data was organized in a custom-made database using Datajoint[55].

**Experiment 1: Characterization of HD tuning in the arena**. We recorded neuronal responses during five 8-min sessions. A first recording session was performed in light (Experiment 1-L0). We then performed the other protocols described below on a moving platform and rotator, before returning the mouse to the same arena and performing three separate 8-min sessions, first in light (Experiment 1-L1), then in darkness (Experiment 1-D), then we repeated a session in light (Experiment 1-L2).

**Experiment 2: Tilt tuning in freely moving animals**. We recorded neural responses when mice walked freely on a platform. Recordings were performed in 5-min blocks during which the setup's axis II and III were fixed. Within a single block with the platform tilted, active locomotion on the platform's surface changed the mice's head azimuth (Az) and tilt orientation (angle γ) together, and these variables are therefore correlated (Supplementary Fig. 4b, yellow, magenta). Rotating the base (rotation along the blue arrow) between blocks added an offset to azimuth, while leaving the range of head tilt unchanged (e.g. Supplementary Fig. 4b, yellow vs. magenta). This manipulation allowed coverage of all possible head azimuth and tilt orientations (plane in Supplementary Fig. 4b), thereby allowing coverage of a large portion of 3D space relatively uniformly (up to α = 60°; Supplementary Fig. 4c).

We perform one additional manipulation of the space covered by the animal: half-way through each block, the platform was rotated using axis I (Supplementary Fig. 4a). This manipulation served as control for the following potential confounding factor: As long as only Axis III is operated, then local azimuth on the platform is anchored to gravity, e.g. the same side of the platform is always placed downward. Therefore, if a cell's firing was anchored to the azimuth on the platform itself, and not to the tilt, its response could be misinterpreted as a tilt response. Changing Axis I multiple times within each block randomizes local azimuth relative to tilt, which prevents this potential confound.

We performed 17 blocks (~68 min) with the following organization: (1) one block where the platform was horizontal (duration: 8 min), (2) eight blocks where the platform was tilted 45° and the base was rotated in steps of 45° (duration: 2.5 min) and (3) eight blocks where the platform was tilted 65° and the base was

rotated in steps of 45° (duration: 5 min). Note that mice tended to upright their head, therefore tilting the mesh 45° and 70° resulted in average head tilts of ~35° and 60°, respectively (e.g. Supplementary Fig. 4c). Together, these 13 blocks allow a relatively uniform sampling of 3D head orientation (at tilt angles up to ~60°) while mice were unrestrained and locomoting freely.

To ensure that tilt space was adequately sampled, we computed the occupancy distribution $d$ (i.e. the time spent) across 73 tilt positions (uniformly distributed in tilt space for up to 60°). Next, we computed the entropy of $d$ $E(d) = -\Sigma p(d).\log_2(p(d))$, ranging from $\log_2(73) = 6.19$ (uniform distribution) to 0 (if the mouse occupies a single point). We excluded cells where $E(d) < 5.6$, which corresponds to mice sampling less than 2/3 of the tilt space. 45% of recorded cells (not counted in Supplementary Table 1) were excluded based on this criterion.

**Experiment 3: Three-dimensional tuning in the rotator**. The rotator was programmed to scan 3D rotation space uniformly using preprogrammed trajectories that sample 200 head tilt orientations uniformly (Supplementary Fig. 6b, red; Supplementary Movie 3); the distance between adjacent points being ~15°. We computed four distinct trajectories (no overlap, Supplementary Fig. 6c, different colors), each of which visited all points once, and in different order. Trajectories traveled through each point in a straight line at a constant velocity (30°/s) and changed direction between points (Supplementary Fig. 6b, c). All trajectories were replayed forward and backward. This technique ensures that the 2D space of head tilt is covered uniformly. While the desired head tilt is achieved by controlling the two innermost axes (I and II), azimuth is varied by rotating axis III (outer) of the rotator at a constant velocity (±15°/s; Supplementary Fig. 6d, red; the velocity is reversed every four rotations). During the trajectory, mice always faced at least 90° away from the second axis (black in Supplementary Fig. 6a) to ensure that the visual field in front of the mouse is not obstructed. We performed the following variants of the protocol: (i) with the LED stripes (placed inside the visual enclosure) on (Experiment 3-L), (ii) off (Experiment 3-D), and (iii) LED on, after the rotator and the visual enclosure were tilted en bloc 60° relative to vertical by operating Axis IV (Supplementary Fig. 6) (Experiment 3-T).

**Experiment 4: Yaw/pitch/roll rotations**. The rotator was programmed to rotate each mouse back and forth in yaw, pitch or roll at a constant velocity of 30°/s. Starting from a velocity of 0°, each movement included an acceleration phase of 1 s to 30°/s, then 380° of rotation at constant velocity and finally a deceleration period of 1 s. To exclude any potential response to accelerations or decelerations, only data recorded during the central 360° of constant-velocity rotation period was used in the analysis.

**Data analysis**. All well isolated neurons recorded during at least one foraging session in light in the arena (Experiment 1-L0, L1 or L2), and during Experiment 2 or Experiment 3-L have been included for analyses, with the following exceptions:

- Recordings in animals H51M, H54M and H59M were performed in an early version of the rotator where the vertical LED stripes and black bands were absent. Cells never exhibited azimuth tuning when recorded in this setup, but could otherwise be classified as azimuth-tuned based on Experiment 1. These animals were excluded from all analyses, except in Supplementary Fig. 18.
- We designed a coverage criteria in Experiment 2, as described above. Neurons that did not pass this criteria were still considered for analysis in Supplementary Fig. 19 only. The corresponding number of neurons are shown in Supplementary Table 1.

We first classified neurons as azimuth-tuned or non-azimuth-tuned based on their responses in the freely moving arena. Neurons could also be classified as azimuth-tuned or azimuth-untuned based on their responses in the platform and rotator. However, because azimuth responses have lower amplitude in the rotator, they often did not reach significance level. Therefore, throughout the study, azimuth-tuned refers by default to the classification based on freely moving data in the arena (Experiment 1).

Similarly, neurons were classified twice as tilt-tuned or not, based on recordings on the orientable platform and in the rotator independently. As with azimuth tuning, neurons that exhibited significant tilt tuning when moving freely may not be significantly tuned in the rotator, because responses in the rotator had lower amplitude. On the contrary, some neurons that exhibited significant tilt tuning in the rotator were not significantly tuned when moving freely because this protocol sampled a limited range (~1/3) of head tilt. Nevertheless, differences were small, and the majority of neurons that were significantly tilt-tuned in one setup were also tuned to the other (Fig. 2).

Importantly, we confirmed that azimuth and tilt tuning in the rotator and freely moving were correlated in terms of amplitude and consistent in terms of spatial characteristics for neurons that were significantly tuned to azimuth or tilt in both experiments (Fig. 2c, d; Supplementary Fig. 8).

For each recorded neuron, we computed the following tuning curves:

(1) To evaluate azimuth tuning, we computed 1D azimuth tuning curves in all conditions of Experiment 1, in Experiment 2 when the platform is horizontal, in Experiment 4-Yaw; and in data points where head tilt was less than 45° during Experiment 3-L.

(2) To evaluate tilt tuning for Experiment 2 and Experiment 3-L,D,T, data were averaged across azimuth. We also computed pitch/roll tuning curves based on Experiment 4.

Note that tilt tuning is different from a recent finding that ADN HD cells encode azimuth in a tilt-dependent manner[13,14], which has been explained by a framework called the dual-axis rule or tilted azimuth (TA); see Supplementary Fig. 9. Although tilt is used to compute TA, TA signals do not carry any information about head tilt since, given any value of TA, all head tilts are still possible. Reciprocally, the tilt tuning identified here does not carry any information about TA since all azimuth are still possible. Thus, 2D tilt tuning and 1D tilted azimuth encode different dimensions of 3D head orientation (see also Supplementary Movie 1).

**Coordinates used to encode 3D head orientation**. For data analysis and fitting purposes (e.g. Supplementary Fig. 10), tilt was expressed in Cartesian coordinates $(G_X, G_Y, G_Z)$ that represent the orientation of the gravity vector (normalized to a length of 1) in head coordinates. Head tilt was also expressed in spherical coordinates to describe tilt tuning curves ($\alpha, \gamma$; Supplementary Fig. 5), where $\alpha$ is the tilt angle: $\alpha = 0°$ in upright orientation (UP) and $\alpha = 180°$ in upside-down orientation (UD); and $\gamma$ encodes tilt orientation: $\gamma = 0°$ and $\gamma = 180°$ correspond to nose-down (ND) and nose-up (NU) tilt (pitch); $\gamma = 90°$ and $\gamma = -90°$ correspond to left ear-down (LED) and right-ear-down (RED) tilt (roll). Spherical coordinates are transformed into Cartesian coordinates and vice-versa by $G_X = \sin(\alpha).\cos(\gamma)$; $G_Y = \sin(\alpha).\sin(\gamma)$; $G_Z = -\cos(\alpha)$; and $\alpha = \mathrm{acos}(-G_Z)$; $\gamma = \mathrm{atan2}(G_Y, G_X)$. Note that Cartesian coordinates and spherical coordinates can be indifferently used to compute tilt tuning curves, as long as spherical distances are computed correctly (e.g. the distance between 170° tilt nose-down, i.e. [$\alpha, \gamma$]= [0, 170] and 170° tilt nose-up, i.e. [$\alpha, \gamma$]= [180,170], is 20°). However, the Cartesian coordinate system, which is more general, allowed modeling of seemingly bimodal tuning curves (in spherical coordinates) using unimodal tuning functions (Supplementary Fig. 10).

Neuronal responses were evaluated by computing tuning curves, which were smoothed using Gaussian kernels with standard deviation of 15° on both azimuth and tilt (we used an equivalent standard deviation of sin(15°) when tilt is expressed in Cartesian coordinates). We computed 3D azimuth tuning curves in 3 different ways, by expressing azimuth in a yaw-only (YO), an earth-horizontal (EH) or a tilted (TA) frame, and found that the latter accounted for neuronal responses better (Fig. 3; Supplementary Fig. 9). Earth-horizontal azimuth is computed by defining a forward pointing vector **N**, aligned with the head's naso-occipital axis, and encoding its orientation in an earth-fixed reference frame $(i, j, k)$, i.e. $\mathbf{N} = (N_i, N_j, N_k)$. Earth-horizontal azimuth is defined as the orientation of N on the earth-horizontal $(I, j)$ plane, i.e. $\mathrm{EHAz} = \mathrm{atan2}(N_j, N_i)$. EH azimuth can be transformed into tilted azimuth by the following equation:

$$\mathrm{TA} = \mathrm{EHAz} - \gamma - \mathrm{atan2}(-\sin(\gamma), \cos(\alpha) \cdot \cos(\gamma)) \quad (1)$$

**Tuning curve fitting**. To quantify tuning curves, von Mises and/or Gaussian functions were fitted and standard shuffling analysis was used to evaluate the statistical significance of azimuth and tilt tuning. 2D tilt tuning curves were fitted with Gaussian distributions (Fig. 4; Supplementary Fig. 10), where tilt was expressed in Cartesian coordinates and

$$\mathrm{FR}_{\mathrm{Ti}}(\alpha, \gamma) = \mathrm{FR}_0 + A.N_{\mathbf{M},\mathbf{C}}(G_X, G_Y, G_Z) \quad (2)$$

where $N_{\mathbf{M},\mathbf{C}}(G_X, G_Y, G_Z)$ is a 3D Gaussian distribution centered on M and with covariance matrix C.

Azimuth tuning curves were fitted with circular normal (von Mises) distributions. Preliminary analysis revealed that the PD of azimuth tuning is maintained when the head tilts (when azimuth is expressed in a tilted frame) but that its gain changes. To account for this, we defined a tilt-dependent gain $g(\alpha)$ and expressed azimuth tuning as:

$$\mathrm{FR}_{\mathrm{Az}}(\mathrm{Az}, \alpha) = g(\alpha) \cdot \exp(K \cdot \cos(\mathrm{Az} - \mathrm{PD}))/\mathrm{l} + (1 - g(\alpha)) \quad (3)$$

where $\kappa$ is the parameter of the van Mises distribution. For convenience, we normalized $\mathrm{FR}_{\mathrm{Az}}(\mathrm{Az}, \alpha)$ such that its average value across all azimuths is 1 (by setting l to the average value of $\exp(\kappa.\cos(\mathrm{Az} - \mathrm{PD}))$).

Finally, we evaluated the interaction between azimuth and tilt tuning by fitted 3D tuning curves defined as the product of the azimuth and tilt tuning cures defined above, i.e.

$$\mathrm{FR}(\mathrm{Az}, \alpha, \gamma) = \mathrm{FR}_{\mathrm{Az}}(\mathrm{Az}, \alpha) \cdot \mathrm{FR}_{\mathrm{Ti}}(\alpha, \gamma) \quad (4)$$

Tilt tuning curves had 11 free parameters: $\mathrm{FR}_0$, $A$, **M** (3-dimensional) and **C** (a covariance matrix, i.e. 6-dimensional). The normalized azimuth tuning curves had 2 free parameters ($\kappa$ and PD). The tilt-dependent gain $g(\alpha)$ was fitted independently at 13 tilt angles $\alpha$ ranging from 0 to 180° by increments of 15°, resulting in 13 additional free parameters. The 3D tuning curves were computed from experimental data at 184 uniformly distributed tilt orientations and 24 azimuth orientations, i.e. 4416 points, and fitted to the 3D curve model by gradient ascent (Matlab function lsqnonlin). Note that since the average value (across all azimuths) of $\mathrm{FR}_{\mathrm{Az}}(\mathrm{Az}, \alpha)$ was 1, the average tilt tuning curve (across all azimuth) was $\mathrm{FR}_{\mathrm{Ti}}(\alpha, \gamma)$.

We tested an additional model that assumes that azimuth and tilt tuning interact additively, i.e.:

$$FR(Az, \alpha, \gamma) = FR_{Az}(Az, \alpha) + FR_{Ti}(\alpha, \gamma) \quad (5)$$

We found that this model did not fit 3D tuning curves as well as the multiplicative model: its correlation coefficient was significantly lower in 33/53 conjunctive cells, and better only in 1/53 conjunctive cell (Fisher $r$ to $z$ transform, at $p < 0.01$), and significantly lower at the population level (median $\rho = 0.85$, [0.80–0.87] CI vs. 0.88, [0.86–0.91] CI, $p < 10^{-8}$, paired Wilcoxon test). Therefore, we used the multiplicative model to model 3D responses in this study.

The length of the mean vector $|\mathbf{R}|$ (i.e. the normalized Rayleigh vector), ranging from 0 to 1, is used commonly to assess how strongly a cell is tuned ($|\mathbf{R}| = 0$, untuned cell; $|\mathbf{R}| = 1$, maximally tuned cell) independently from its average firing rate. It allows comparing cells with a large range of peak firing rates. Thus, azimuth tuning was quantified consistently with previous studies by computing the mean vector:

$$\mathbf{R} = c \cdot \sum FR(Az) \times \exp(-i \times Az) / \sum FR(Az) \quad (6)$$

where FR(Az) was sampled at 100 positions separated by 3.6° and $c = 3.6 \times \pi/180/2/\sin(1.8)$[56].

Mean vectors can be generalized to a 2D distribution by expressing tilt in Cartesian coordinates $\mathbf{G} = (G_X, G_Y, G_Z)$ and computing:

$$\mathbf{R} = \sum FR(G) \times G / \sum FR(G) \quad (7)$$

The resulting 2D vector has a length of 1 if all spikes occur at the same tilt and 0 if spikes are distributed uniformly or symmetrically. However, because mean vectors computed in 1D and 2D cannot be compared directly, we developed an alternative measure called normalized tuning amplitude (NTA; Supplementary Fig. 7). The normalized tuning amplitude of an 1D azimuth or 2D tilt tuning curve was defined based on the maximum ($FR_{max}$) and minimum ($FR_0$) firing rate, as $NTA = (FR_{max} - FR_0)/FR_{max}$. Thus, normalized tuning amplitude ranged from 1 (when a tuning curve ranged from 0 Hz to a peak value) to 0 (when a cell was unmodulated). Note that normalized tuning amplitude measures the cell's modulation amplitude, but not the sharpness of the tuning curve.

We defined a cell's tilt preferred direction (PD) as the orientation at which the fitted 2D tilt tuning curve is maximal. In cells with bimodal tilt tuning (Supplementary Fig. 10), the PD corresponded to the highest peak. We compared the differences between PD recorded in light and darkness in Fig. 5d. If a bimodal cell has two peaks with approximately equal amplitude, the highest peak measured in light may become smaller in darkness, resulting in an apparent change in spatial tuning. To prevent this, we compared the PD in light with the direction of the two peaks measured in darkness, and the difference in PD was defined as the smallest difference.

**Statistical procedures to determine significant tuning.** We used a shuffling procedure[57,58] to assess the statistical significance of azimuth or tilt tuning. Each sample was generated by (1) shifting the entire spike train circularly by a random value of at least ±10 s, (2) recomputing the tuning curve, (3) performing the Gaussian fit and (4) computing the azimuth and/or tilt normalized tuning amplitude. We computed the mean value $m$ and standard deviation $\sigma$ of the normalized tuning amplitude across 100 shuffled samples. The statistical $p$-value of the normalized tuning amplitude NTA measured in the un-shuffled data was computed as $1 - F(NTA, m, \sigma)$, where $F$ is the cumulative Gaussian distribution with average $m$ and standard deviation $\sigma$. The standard deviation of the model's correlation across all samples is also used as an estimate of the standard error of the model's correlation.

We considered azimuth or tilt tuning to be significant if (1) the $p$-value computed as described above was less than 0.01 and (2) the normalized tuning amplitude NTA was equal to or higher than 0.25. The second criterion was equivalent to selecting cells where the modulation was at least one third of the baseline firing rate, and was used to eliminate cells with very small but significant modulation (how this threshold compares to criteria used in other studies is analyzed and discussed in Supplementary Fig. 7).

We combined data from multiple repetitions of Experiment 1-L to assess if cells were significantly tuned to azimuth when the animal was moving freely. We used two techniques to combine multiple repetitions: (1) we analyzed each repetition independently and computed the median $p$-value and amplitude across repetitions, and (2) we computed a $p$-value and amplitude based on data pooled across all repetitions. We tested if the values obtained with technique (1) or with technique (2) passed the criteria described above, and classified cells as azimuth-tuned if they passed any of these tests. This two-technique approach was used because pooling data yields a greater statistical power but fails if the cell's PD shifted between sessions, whereas the second approach is not affected by shifts in the PD. In total, 37% cells passed both tests; 6.5% passed the first test only, 7.5% passed the second test only; therefore, in total, 51% cells passed one or the other and were classified as azimuth-tuned.

We used data from Experiment 3-L to assess if cells are significantly tuned to tilt, by testing if the normalized tuning amplitude of the average tilt tuning curve (across all azimuths) passed the criteria described above. We also tested if cells are significantly tuned to azimuth in the rotator by computing the azimuth tuning

curve based on data for up to 45° tilt and testing if its normalized tuning amplitude passes the criteria described above. In some cells, Experiment 2 or Experiment 3-L were repeated multiple times. We found that the preferred direction of tilt and azimuth tuning were stable across repetitions, and pooled data across all repetitions.

**Partial correlation of azimuth tuning.** We compared models of azimuth tuning (e.g. in Fig. 6g; Supplementary Fig. 11) by fitting 3D neuronal responses with 3D models that included the respective azimuth models and comparing the models' coefficient of correlation. However, in most cells, this correlation is largely determined by the neuron's tilt tuning. To better reveal the specific contribution of azimuth responses, we computed the partial correlation specifically attributable to azimuth tuning by removing the contribution of gravity. We first fitted the 3D model normally and computed the sum of squared residuals ssr. Next, we removed azimuth tuning from the best-fitting model (by setting $g(\alpha)$ to zero) to create a tilt-only model, and computed the sum of squared residuals $ssr^G$. We defined the partial correlation coefficient of azimuth as $\rho = sqrt((ssr^G - ssr)/ssr^G)$. Intuitively, this coefficient measures how much the tilt-only model's error is reduced by the addition of azimuth tuning.

**Tilt and azimuth velocity analysis.** We performed another Gaussian curve fitting to test whether neurons carry a mixture of tilt and tilt derivative information. We expressed head tilt measured during Experiment 3-L in Cartesian coordinates ($G_X$, $G_Y$, $G_Z$) and then computed the time derivative of the gravity vector ($dG_X/dt$, $dG_Y/dt$, $dG_Z/dt$) as a measure of tilt velocity. Next, we fitted neuronal firing rate with $FR = FR_0 + A.N_{\mathbf{M,C}}(G_X, G_Y, G_Z) + A'.N_{\mathbf{M',C}}(dG_X/dt, dG_Y/dt, dG_Z/dt)$. We computed the normalized tuning amplitude of tilt and tilt velocity and used the same shuffling method and criterion ($p < 0.01$, NTA > 0.25) as in other Gaussian fits to assess whether cells were significantly tuned to the gravity derivative. Note that the tilt tuning curves obtained with this method were identical to those obtained when fitting the 3D model that includes tilt and azimuth tuning.

We investigated whether cells encode azimuth velocity (dAz/dt) by computing azimuth velocity tuning curves for each cell (i.e. average firing rate as a function of dAz/dt) using all data from Experiment 1-L. The tuning curves were evaluated at all velocities ranging from −200 to 200°/s by increment of 20°/s and smoothed using a 8°/s Gaussian kernel. We computed the amplitude and normalized tuning amplitude of these curves, and used the same shuffling method and criterion ($p < 0.01$, NTA > 0.25) to assess whether cells were significantly tuned.

**Experiment 3-T.** To analyze the results of Experiment 3T, we expressed 3D head motion (tilt and azimuth) in both gravity and visual reference frame. Motion in a gravity frame was computed based on the actual 3D position of the head, and decomposed into gravity-referenced tilt (**G**, in Cartesian coordinates) and tilted azimuth ($TA^G$). Motion in a visual frame was computed as if Axis IV of the system had not been tilted, and decomposed into visually referenced tilt (**V**, in Cartesian coordinates) and tilted azimuth ($TA^V$). Next, we assumed that HD cells encode a tilt signal computed as a weighted average of **G** and **V** (**T**($w$), computed as $w.\mathbf{G} + (1-w).\mathbf{V}$ and normalized to a length of 1) and an azimuth signal equal to $TA^G$ or $TA^V$. We computed the cells' 3D tuning curves for each possible value of $w$, ranging from −1 to 2, and based on $TA^G$ or $TA^V$. Each tuning curve was compared to the tuning curve fitted to the data measured with the rotator upright (Experiment 3-L) by computing their pixel-by-pixel correlation. We used the shuffling procedure described above to generate 60 shuffled samples of the tuning curves in Experiment 3-T. The standard deviation of the correlation between the curve fitted to Experiment 3-L and these samples was used as an estimate of the standard error of the correlation, and 99% confidence intervals were set to 2.56 times the standard error. To compare the two models of tilted azimuth ($TA^G$ and $TA^V$), we computed the partial correlation of azimuth tuning as described above.

**Experiment 4P/R.** Using the full 3D tuning curve data from Experiment 3-L, we predicted the responses to pitch and roll rotations by sampling the 3D tuning curve at the head orientations visited during pitch and roll rotations. The pitch and roll tuning curves measured during Experiment 4 and predicted based on Experiment 3 were then fitted with 1D Gaussians, and the resulting modulation amplitudes and preferred direction were compared. Data during Experiment 3 and 4 in light and darkness were averaged.

**Cross-correlation analysis.** We performed a similar analysis as in Peyrache et al.[26] in Supplementary Fig. 19. Neuronal firing rates were sampled in 33 ms time bins and smothered using a Gaussian filter with 300 ms standard deviation. Cross-correlograms between pairs of simultaneously recorded cells were computed using the Matlab function xcoeff with the normalization option "coeff". Finally, the median value of the cross-correlograms for time lag >3 s was subtracted from the cross-correlograms.

**Firing properties.** We confirmed that the distribution of average firing rates and CV2 were similar for all cells in the CIN, ADN, and RSC (Supplementary Fig. 18a).

Furthermore, we observed that most spikes recorded in CIN had small (<0.33 ms) trough to peak durations, indicative of axonal spikes (Supplementary Fig. 18b).

**Reporting Summary**. Further information on research design is available in the Nature Research Reporting Summary linked to this article.

## Data availability

Source data underlying Figs. 2a–d, 3d, g, 4c, 5a–e and 6e–g are provided as a Source Data file. Further data are available from the corresponding author upon reasonable request. A reporting summary for this Article is available as a Supplementary Information file.

## Code availability

Computer code necessary to reproduce the study's conclusion is available from the corresponding authors upon reasonable request.

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

## Acknowledgements
This study was supported by the Simons Collaboration on the Global Brain Grant 542949, NIH R01-AT010459 and R01 DC004260.

## Author contributions
J.L. designed and supervised experiments, analyzed data and wrote manuscript. J.N., A.M.A., H.C. performed experiments and analyzed data. J.D.D. and E.A. supervised experiments. D.E.A. designed and supervised experiments and wrote manuscript.

## Competing interests
The authors declare no competing interests.
