## [Peer Review File · Nature Communications]

Reviewers' comments:

Reviewer #1 (Remarks to the Author):

Three-dimensional (3D) orientation in space is required for many forms of behavior. Angelaki et al. reports neurons tuned to 3D head-orientation in the mouse brain. This work, together with previous studies in bats and monkeys, establishes the existence of neurons acting similarly to a 3D neural compass as a general feature of the mammalian brain. In order to understand the reference frame used by 3D head-direction cells in mice, Angelaki et al. performed an impressive series of manipulations in freely-moving and restrained animals. These experiments demonstrate that tilt-tuned head-direction cells are more anchored to gravity, than to visual cues. Additionally this study provides further insights regarding the interactions between tilt and azimuthal-tuning.

Major Comments

- 1) Throughout the paper tuning to pitch and roll is treated equivalently, despite a clear over-representation of cells that are modulated by pitch. Are there other systematic differences between pitch- and roll-preferring cells? For instance, is roll tuning weaker than pitch tuning, in terms of modulation depth etc? Are roll-preferring cells anchored to gravity as strongly as pitch cells? Are pitch and roll cells equally stable in terms of their PD?
- 2) It would be useful to show 1D pitch and roll tuning in order to allow comparison with other studies.
- 3) Are tilt-tuned cells show a different anatomical organization compared to azimuth-tuned cells - e.g. similarly to the functional-anatomical gradient shown in bats (Finkelstein et al 2015)?
- 4) In Figure 4d a large fraction of cells were not significantly correlated with either gravity or visual reference frame. Could it be that some cells combine gravity signal with visual cues so that their tuning becomes a weighted average of the two reference frames?
Are these cells differ in other aspects as well? For instance, are these cell have a weaker/broader tilt or azimuth tuning? It would also help to show more example cells (in addition to the hypothetical cells shown in Sup Fig14).
- 5) I missed an explicit analysis of tuning unimodality. The authors only mention that: "The majority of 3D tilt tuning curves were unimodal (as in the example responses of Fig. 1k,l, see also Suppl. Fig. 9a). However, bimodal (Suppl. Fig. 9b) or complex-shaped (Suppl. Fig. 9c) tuning curves were also encountered. Regardless of their shape, tilt tuning curves were well fitted with Gaussian functions (Suppl. Fig. 9,10), which allowed quantification of tilt preferred directions in 3D".
What is the proportion of unimodal cells? For multimodal cells, such as Fig. 9b – is there a coordinate system that would explain their tuning better? For example, although I didn't quite follow the explanation of the Gaussian fits, it seems that for some cells these fits (in Cartesian coordinates) result in a unimodal tuning instead of a multimodal tuning (in spherical coordinate). Does this suggest that explaining the coding of these cells in spherical coordinate is inadequate?
- 6) I generally agree with the author's explanation for azimuthal reversal when the head is inverted. However, the following issues require clarification:
In Fig. 5f – azimuth tuning for tilts > 90 deg in pitch is expected to show reversal (as shown for an example cell in Sup Fig 15). This should be true for tilt in pitch, but not for tilt in roll. In Fig. 5f, tilt is not separated

by pitch and roll and it is unclear if this reversal indeed happens. Is azimuth in Fig. 5f expressed in TA frame?

The authors could choose different spherical coordinates to represent tilt - e.g. with pitch spanning 0-360 deg, and roll spanning [0-180 deg]. In this case, a change in azimuth from 0 to 180 deg (when azimuth is defined in earth horizontal frame) could be achieved by a 180 degree rotation in pitch. Therefore, the authors' statement that then the head "would be rolling instead of pitching, and azimuth reversal would not occur" (Page 61 line 1269) depends on the reference frame chosen. The finding in bats (Finkelstein et al 2015) showing that azimuth reversal occurs for pitch >90deg -- including at *180 deg* pitch -- supports this alternative.

Minor Comments

- 1) The following sentence in the Abstract (Line 32) would likely be unclear to a reader not familiar with the terms "head-horizontal plane" and "terrestrial allocentric world": "When the head tilts, azimuth tuning is affixed to the head-horizontal plane, but also uses gravity to remain anchored to the terrestrial allocentric world."
- 2) The cingulum and the retrosplenial cortex (until recently) are not typical areas for recording head-direction cells from. It would be useful to explain the motivation behind recording in these areas in the introduction.
- 3) The "untuned" cell in Fig.1d does seem to have an azimuthal tuning in Fig1.g – two pink hotspots at ~90deg Azimuth. I am surprised this tuning didn't come up in the marginal tuning in Fig.1d.
- 4) Page 4, Line 70: "tilt angle from upright: α (range: 0-180°)" – this a confusing definition. From this definitions it seems that α is equivalent to pitch, yet NU and ND orientations have the same α .
- 5) Fig1. k,l – it would help to make the 3D plots semi-transparent.
- 6) Sup Fig.4 – it is not clear from the schematics what Axis I is. Also the corresponding sentence in the legend is not clear: "Axis I is rotated randomly in the middle of each block to prevent tilt orientation relative to gravity from being coupled to head orientation relative to local cues on the platform itself."
- 7) NTA measurement will give a very different answer from RV if the 2D tuning is not single peaked.
- 8) Sup Fig.5 – the following sentence requires more explanation : "The rationale for the coordinate system is the following: each tilt orientation relative to gravity is equivalent to, and defined as, the corresponding orientation of the gravity vector relative to the head".
- 9) Why in Fig4.d z-scored partial correlation coefficients are used instead of just correlation coefficients?
- 10) Sup Fig.9 line 1160 should be UP, *UD*,RED
- 11) Sup Fig. 13 – Some spike-sorting metrics are required to support the claim that the same cell was recorded over a 2-week period.

Reviewer #2 (Remarks to the Author):

In the present manuscript, Angelaki and collaborators report new data providing evidence for a modulation of the mouse head-direction (HD) system in three dimensions. The data are compelling and support the main claims of the paper. However, one crucial aspect of the manuscript is the representation of the data and, in its present form, the manuscript is sometimes hard to follow.

Major comments

It is sometimes hard to figure out the way the authors represent their data and how it validates and invalidates other models, especially when it comes to azimuth. In fig. 1, it should be made extremely clear what azimuth is, whether it's relative to the head's plane or the earth plane. The first panels in Fig. 1 could be more straightforward, as it is unclear what "vision" refers to in this case.

A lot of neurons seem to be modulated at "nose down" or "nose up" (ND/NU) (Fig. 3). One can think that it is the simple consequence that in these positions the azimuth is undefined (or 'degenerate'). A pure "azimuth-tuned cell" would certainly fire for all azimuth directions in ND or NU position unlike in UP, when the cell is expected to fire for specific azimuth.

It is not immediately clear how to reconcile these results with the hypothesis that the HD system is a 1d ring attractor. The present data seem to suggest otherwise, in particular that the system has additional possible states that are not fully explained by azimuthal tuning. The idea of the tilted azimuth (TA) is apparently that, indeed, the HD system is a 1d ring whose member neurons fire for angles relative to the orientation of the head in 3D. It is necessary to carefully explain the hypothesis here.

The fact that pairwise differences of preferred direction in the azimuthal plane (Fig. S8) are preserved is a good indication that the activity may arise from a 1D ring attractor. However, along the same lines as the previous comment, should the neurons be additionally modulated by tilt, pairwise correlation during sleep (As previously reported) could not be fully explained by their azimuth tuning. Otherwise, if the authors think that these observations are not incompatible, this point should be discussed.

The similarities with the bat HD system should be also carefully discussed since there are many differences. First, in bats, the recordings were performed in the post-subiculum. Furthermore, the tuning curves of the HD cells in the bat system were different from what is shown in the present manuscript. In bats, some cells fire for specific values of azimuth and pitch, for example. It was not the case here. Last, in bats, there is an anatomical gradient of tuning. Is there any sign that such anatomical distribution exists in mice?

Last but not least, the trough-to-peak duration of ADN spikes are very short, comparable to axonal spikes and/or fast-spiking neurons. ADN neurons are pyramidal thalamocortical neurons that should show trough-to-peak duration of at least 300ms. The examples in S1 are also a bit surprising. How come all waveforms are positive except on the highest-amplitude channel? Seems like it results from median or average filtering (i.e. the median/average of all channels is subtracted from each channel to remove noise). Would it be possible to show raw spike waveforms? Are spikes usually detected on more than

one channel (axonal spikes are usually not too widespread)?

Minor comments.

“In fact, 76/128 cells (14/22 ADN; 15/27 RSC; 47/79 CIN) were significantly ($p < 0.01$) more correlated to the gravity reference frame than the visual frame, and the remaining 52 cells were not significantly more correlated in either frame.” This is pretty unclear. What are these 52 cells correlated to then?

The authors use several tuning characterization. It is unclear what the “Azimuth tuning amplitude” of Fig 2 corresponds to: peak firing rate only or relative to baseline? What do the plots in Fig. 2 then show, expect that firing rate are overall preserved but slightly modulated by the conditions. Computing information per spike for each condition would have perhaps been more informative.

We thank the reviewers for their constructive comments. In response to their remarks, we have performed some major revisions to the manuscript, including in particular:

- We have recorded neuronal responses in the dysgranular RSC of two additional animals to better compare dysgranular and granular RSC (we found no difference in population response).
- We have included responses recorded in the ADN of three animals using an earlier version of the 3D rotation protocol in **Suppl. Fig. 18** to better characterize the properties of HDC in this region. We are able to classify cells recorded in these animals as conjunctive, tilt-only, azimuth-only, etc, based on this earlier version, which allows us to include them in this figure. However, because of differences in experimental design, we didn't include them in the main figures.
- We have entirely re-analyzed data recorded when the rotator is tilted. We can now confirm that tilt responses are precisely referenced to a gravity-frame. Furthermore, we found that the earth-horizontal plane (or equivalently earth-vertical axis) involved in implementing the dual-axis rule (or TA model) are also defined by gravity.
- We have performed a similar analysis as in Peyrache et al. 2015 (**Suppl. Fig. 19**), using responses recorded on the platform, to investigate if the 1D attractor model is compatible with HDC responses in 3D. We found that attractor activity weakens when the platform is tilted, suggesting that the 1D attractor may exist only when the head is upright.
- We have modified and re-ordered several other figures to answer other specific questions and improve the clarity of the manuscript.

We have included a redlined file where all changes are marked to our re-submission.

Reviewer #1 (Remarks to the Author):

Three-dimensional (3D) orientation in space is required for many forms of behavior. Angelaki et al. reports neurons tuned to 3D head-orientation in the mouse brain. This work, together with previous studies in bats and monkeys, establishes the existence of neurons acting similarly to a 3D neural compass as a general feature of the mammalian brain. In order to understand the reference frame used by 3D head-direction cells in mice, Angelaki et al. performed an impressive series of manipulations in freely-moving and restrained animals. These experiments demonstrate that tilt-tuned head-direction cells are more anchored to gravity, than to visual cues. Additionally this study provides further insights regarding the interactions between tilt and azimuthal-tuning.

Major Comments

1) Throughout the paper tuning to pitch and roll is treated equivalently, despite a clear over-representation of cells that are modulated by pitch. Are there other systematic differences between pitch- and roll-preferring cells? For instance, is roll tuning weaker than pitch tuning, in terms of modulation depth etc? Are roll-preferring cells anchored to gravity as strongly as pitch cells? Are pitch and roll cells equally stable in terms of their PD?

*We have investigated this question and we found that cells tuned in the roll plane have lower firing rate (**Fig. 5b**). Yet, other properties are unchanged, including normalized tuning amplitude (**Fig. 5c**), anchoring to gravity (**Fig. 6d**, we mentioned it in the main text) and stability (**Fig. S14h**, we mentioned it in the legend).*

Our findings confirm the pre-eminence of cells tuned to pitch movement (also found in other studies). Yet, roll-tuned cells do exist and must be accounted for. Furthermore, from a modeling point of view, we seek to understand spatial orientation when moving in 3D, which includes roll movement. Our model must account for the response of pitch-tuned cells during roll (as well as the response of azimuth cells; after all the dual-axis rule would make little point if the head never tilted in roll). Therefore, our theoretical framework must include all three dimensions.

2) It would be useful to show 1D pitch and roll tuning in order to allow comparison with other studies.

*We show it in **Suppl. Fig. 13**, which we extended to add an example cell. This analysis confirmed that gravity tuning is similar during 1D pitch and roll rotations and in our 3D rotation protocol. We also dedicated a figure to comparing our results with those of Shinder and Taube (2019) (we also discuss this paper in detail in another publication, Laurens and Angelaki 2019).*

3) Are tilt-tuned cells show a different anatomical organization compared to azimuth-tuned cells - e.g. similarly to the functional-anatomical gradient shown in bats (Finkelstein et al 2015)?

*We have investigated this question (in **Suppl. Fig. 18**) and didn't find any clear anatomical gradient within the ADN, nor any difference between the granular and dysgranular RSC.*

4) In Figure 4d a large fraction of cells were not significantly correlated with either gravity or visual reference frame. Could it be that some cells combine gravity signal with visual cues so that their tuning becomes a weighted average of the two reference frames?

*This is a very good point. We re-designed our analysis of this protocol entirely to address it: we now express tilt as a weighted average of the two frames, and determine the best fitting weight on a cell-by-cell basis. We find that the population weight is exactly centered on a gravity weight of 1 (**Fig. 6**). Furthermore, no cell correlates significantly better with a visually-referenced or intermediate frame (except one, likely a false positive) compared to the gravitational frame (**Suppl. Fig. 15l,m**). We thank the reviewer for suggesting this!*

Are these cells differ in other aspects as well? For instance, are these cells have a weaker/broader tilt or azimuth tuning? It would also help to show more example cells (in addition to the hypothetical cells shown in Sup Fig14 [note from the authors: these hypothetical cells are not shown any more]).

*These are typically cells with weaker tuning (to be specific, they have lower peak firing rates and as a consequence low peak-to-trough amplitude). As a consequence, these cells are comparatively noisier, such that the correlation in both frames is lower and doesn't pass the significance threshold on a cell-by-cell basis (even though, at the population level, most of these cells are below the diagonal, i.e. correlate better with gravity). We show an example cell in **Suppl. Fig. 15n**.*

5) I missed an explicit analysis of tuning unimodality. The authors only mention that: "The majority of 3D tilt tuning curves were unimodal (as in the example responses of Fig. 1k,l, see also Suppl. Fig. 9a). However, bimodal (Suppl. Fig. 9b) or complex-shaped (Suppl. Fig. 9c) tuning curves were also encountered. Regardless of their shape, tilt tuning curves were well fitted with Gaussian functions (Suppl. Fig. 9,10), which allowed quantification of tilt preferred directions in 3D".

What is the proportion of unimodal cells? For multimodal cells, such as Fig. 9b – is there a coordinate system that would explain their tuning better? For example, although I didn't quite follow the explanation of the Gaussian fits, it seems that for some cells these fits (in Cartesian coordinates) result in a unimodal tuning instead of a multimodal tuning (in spherical coordinate). Does this suggest that explaining the coding of these cells in spherical coordinate is inadequate?

Yes, exactly! To be more specific, we think that the 3D Cartesian space is the simplest to model the tilt tuning of the cells. Once the model is fit, it is still possible to restrict the 3D space to a sphere (since encoding head tilt relative to vertical only requires using a spherical subspace) and to express the tuning curves in 2D spherical coordinates, as we do (because it makes it much simpler to describe a given head orientation).

Note that all the data analysis is done in Cartesian coordinates. We only used a spherical coordinates system (the α , γ system in **Suppl. Fig. 5**) to describe particular head orientations, because it is clearer to say “the cell fires maximally when the head tilts $\alpha=60^\circ$ towards nose-down ($\gamma=0^\circ$)” rather than “the cell fires maximally when $[GX, GY, GZ]=[0.86, 0, -0.5]$ ”.

From a mechanistic point of view, we can propose a simple explanation: gravity is sensed in 3D Cartesian coordinates by the otoliths in the inner ear, and these signals could be filtered through 3D Gaussian tuning functions. We now mention this possibility in discussion: at this point it is of course hypothetical, yet it is a relatively straightforward explanation for our findings.

Note also that it would be ok to compute tuning curves in spherical coordinates, as long as one computes spherical distances correctly (e.g. the distance between 170° tilt nose-down, i.e. $[\alpha, \gamma]=[0, 170]$ and 170° tilt nose-up, i.e. $[\alpha, \gamma]=[180, 170]$, is 20°).

*We improved the explanation of Gaussian tuning and placed it in a main figure (**Fig. 4**). We also performed an analysis of bimodality (**Suppl. Fig. 10**).*

6) I generally agree with the author's explanation for azimuthal reversal when the head is inverted. However, the following issues require clarification:

In Fig. 5f – azimuth tuning for tilts $>90^\circ$ in pitch is expected to show reversal (as shown for an example cell in Sup Fig 15). This should be true for tilt in pitch, but not for tilt in roll. In Fig. 5f, tilt is not separated by pitch and roll and it is unclear if this reversal indeed happens. Is azimuth in Fig. 5f expressed in TA frame? [Note from the authors: Fig. 5 and Suppl. Fig. 15 are now Fig. 3 and Suppl. Fig. 9].

*Yes, the azimuth is expressed in TA frame in **Fig. 3f** (we clarified this in the axis label). Since the reversal during pitch occurs only when tilted azimuth is expressed in an earth-horizontal frame, it is not an object here.*

*Note that we did indeed observe that azimuth reverses during pitch, and not roll, when expressed in an earth-horizontal frame (**Suppl. Fig. 9d**, left panel).*

The authors could choose different spherical coordinates to represent tilt - e.g. with pitch spanning 0-360 deg, and roll spanning [0-180 deg]. In this case, a change in azimuth from 0 to 180 deg (when azimuth is defined in earth horizontal frame) could be achieved by a 180 degree rotation in pitch. Therefore, the authors' statement that then the head “would be rolling instead of pitching, and azimuth reversal would not occur” (Page 61 line 1269) depends on the reference frame chosen [note from the

authors: this refers to Suppl. Fig. 9a]. The finding in bats (Finkelstein et al 2015) showing that azimuth reversal occurs for pitch >90deg -- including at *180 deg* pitch -- supports this alternative.

The reviewer proposes that the brain uses a special rule to compute azimuth when the head is inverted (instead of using the general definition of the TA frame which has a singularity point at this orientation). Indeed, this can explain Ulanovsky's data (who found a consistent tuning in inverted bats). We are, a priori, not opposed to this notion and we discuss it the discussion.

In our opinion, what exactly happens in inverted orientation is still unresolved. We think that additional studies in inverted animals (in particular freely moving animals, such that azimuth responses would be stronger than in our experiments) would be desirable and hope that additional data will be available in the future.

*In **Suppl. Fig. 9**, our purpose here is to explain that the 'normal' definition of tilted azimuth leads to a singularity point in inverted orientation, where azimuth can't be computed unambiguously (this singularity also described in Page et al. 2017). We think that it is important to make this point and therefore we still mention it in the legend.*

Minor Comments

1) The following sentence in the Abstract (Line 32) would likely be unclear to a reader not familiar with the terms "head-horizontal plane" and "terrestrial allocentric world": "When the head tilts, azimuth tuning is affixed to the head-horizontal plane, but also uses gravity to remain anchored to the terrestrial allocentric world."

We reformulated to use a clearer - and still concise – description.

2) The cingulum and the retrosplenial cortex (until recently) are not typical areas for recording head-direction cells from. It would be useful to explain the motivation behind recording in these areas in the introduction.

We are now discussing these questions (together with the presence of 3D HDC in the presubiculum, as shown by Finkelstein et al. 2015) in the section "Three-dimensional HDC across brain regions".

*In our opinion, the retrosplenial cortex ***is*** a typical area for recording head direction cells – thanks to recent studies, indeed. It is true that HD responses are somewhat scarcer and weaker in this area compare to the ADN, and many cells are likely multisensory, see our manuscript <https://doi.org/10.1101/684464> as well as <https://doi.org/10.1101/702712>. Yet, this doesn't mean to us that HDC responses in the RSC are not worth studying. In fact (as shown e.g. by Jacob et al. 2017), the RSC is possibly a major entry point of visual landmark information in the HDC network (it is also generally thought to be involved in egocentric/allocentric transformations which is a related process). At some point, this will raise the question of whether the RSC performs these transformations in 3D. Even though we are not yet able to tackle this question, it seems to us that laying the groundwork by showing that HDC in the RSC are tuned in 3D is an important first step.*

Regarding the cingulum fiber bundle, we were motivated by addressing this possible argument: tilt tuning in ADN and RSC could be a form of ‘internal variable’ in these regions, used for instance for implementing the dual-axis rule, but that would not be part of a 3D orientation signal. 3D-tuned HDC would then simply be part of a circuitry in their regions but wouldn’t project outside of it. Recording in the cingulum offers a good way to address this argument, since it likely carries mainly RSC and anterior thalamus projections at the level at which we recorded it, as indicated by anatomical studies (recently reviewed by Bubb and Aggleton) as well as our own recordings: see <https://doi.org/10.1101/684464>). We think that demonstrating that HDC in the cingulum encode 3D orientation provides a compelling argument that the head direction signal represented and transmitted by the ADN and RSC is inherently three-dimensional.

3) The “untuned” cell in Fig.1d does seem to have an azimuthal tuning in Fig1.g – two pink hotspots at ~90deg Azimuth. I am surprised this tuning didn’t come up in the marginal tuning in Fig.1d.

The marginal tuning in Fig. 1d is based on data on the horizontal platform (which would appear at the bottom of the tuning curve). One can see how the marginal azimuth tuning curve evolves as a function of tilt angle in Suppl. Movie 2. Indeed, it acquires an azimuth tuning at 60° tilt which is weak (NTA=0.34, Rayleigh vector = 0.1) but technically passes the criterion of NTA>0.25). It is possible that this cell has a weak azimuth tuning that can be seen during NU tilt (the cell’s preferred tilt) but not when upright (e.g. in the arena). We could confirm this if we had access to a larger range of tilt, or if the azimuth tuning was strong enough to be visible in the rotator.

This points to one limitation of our study – the neuronal responses measured in the full 3D space in the rotator are attenuated since animals are restrained. Although responses are qualitatively similar in restrained and freely moving animals, we may miss some subtle responses that would be visible in full 3D curves in freely moving animals. To date, it is not easy though to measure uniformly-spaced 3D tuning in rodents.

4) Page 4, Line 70: “” – this a confusing definition. From this definition it seems that α is equivalent to pitch, yet NU and ND orientations have the same α .

We clarified the definition (α refers to the tilt angle in the pitch or roll plane).

5) Fig1. k,l – it would help to make the 3D plots semi-transparent.

We attempted to create semi-transparent plots but found the results quite illegible as soon as the 3D tuning curves are more complex than a single ‘hotspot’ of activity. Instead, we have created animated movies which we think provides a more informative content.

6) Sup Fig.4 – it is not clear from the schematics what Axis I is. Also the corresponding sentence in the legend is not clear: “Axis I is rotated randomly in the middle of each block to prevent tilt orientation relative to gravity from being coupled to head orientation relative to local cues on the platform itself.”

We re-drew Axis I and expanded on the explanation of why we used it (it is to avoid a potential confounding factor).

7) NTA measurement will give a very different answer from RV if the 2D tuning is not single peaked.

*Absolutely; the RV will be close to zero if the 2D tuning is double peaked, or for double-peaked HD tuning curves. This is one of the reasons we don't use it to quantify tilt tuning. Another reason is that the RV is disproportionately affected by the cell's baseline firing rate. For instance, the example tuning curve in **Suppl. Fig. 7a** has a RV of 0.37, which would not pass the criterion of $RV > 0.4$ in some studies! And simply removing the cell's baseline firing would cause the RV to increase to 0.91, which sounds disproportionate. In our opinion, the RV is useful for separating highly tuned HDC (such as classical ADN HDC) from other cells, but isn't well suited for quantifying cells with more subtle forms of HD tuning. We find the NTA to be a more useful measure, that can be applied to several type of motion variables (we used it to quantify place cells, theta-rhythm modulation, etc, in <https://doi.org/10.1101/684464>). We dedicated **Suppl. Fig. 7** to discussing the use of NTA for classifying HDC, and we added a mention to the case of double-peaked curves to the legend.*

8) Sup Fig.5 – the following sentence requires more explanation : “The rationale for the coordinate system is the following: each tilt orientation relative to gravity is equivalent to, and defined as, the corresponding orientation of the gravity vector relative to the head”.

We explained this part with more details.

9) Why in Fig4.d z-scored partial correlation coefficients are used instead of just correlation coefficients?

We were aiming at better showing the difference between the two models (using partial coefficients of correlation), and we used a z-score analysis for statistical purposes. We have now developed an entirely new analysis (following the reviewer's suggestion); where statistical analyses are based on permutation tests.

10) Sup Fig.9 line 1160 should be UP, *UD*,RED

This figure has been redesigned.

11) Sup Fig. 13 – Some spike-sorting metrics are required to support the claim that the same cell was recorded over a 2-week period.

We added an analysis of the inter-spike interval to support this claim. Note that the tetrode was not advanced in this period.

*We would also like to clarify that our criteria to track a cell across multiple recordings within a single day were based on spike sorting alone (such that we could compare different experimental conditions in an un-biased fashion). In a separate step, we merged neurons recorded on different days when spiking characteristics and 3D tuning were identical (this occurred in 91/549 neurons). From a statistical point of view, merging these neurons could not bias our results (outside of **Suppl. Fig. 14**) since we simply merged identical data points (that most likely originate from the same neuron).*

*Therefore, **Suppl. Fig. 14** does not demonstrate that 3D HD tuning is stable in an unbiased fashion; instead it shows that 3D tuning can be stable over several repetitions (we performed this analysis at the request of one of our colleagues). We clarified these points carefully in the legend of the figure and in the Methods: section “Spike sorting and identification of neurons across multiple recording sessions”. Performing an unbiased analysis of neuronal stability over*

days would require the ability to identify neurons across days even if their 3D tuning varies from day to day, which could be accomplished with optical imaging but, to our knowledge, not with tetrodes.

Reviewer #2 (Remarks to the Author):

In the present manuscript, Angelaki and collaborators report new data providing evidence for a modulation of the mouse head-direction (HD) system in three dimensions. The data are compelling and support the main claims of the paper. However, one crucial aspect of the manuscript is the representation of the data and, in its present form, the manuscript is sometimes hard to follow.

Major comments

It is sometimes hard to figure out the way the authors represent their data and how it validates and invalidates other models, especially when it comes to azimuth. In fig. 1, it should be made extremely clear what azimuth is, whether it's relative to the head's plane or the earth plane. The first panels in Fig. 1 could be more straightforward, as it is unclear what "vision" refers to in this case.

*We re-drew this panel to provide a simple explanation of tilted azimuth, and we made sure that the figure axes are always labelled "tilted azimuth" when they show it. We explain the tilt coordinates as clearly as possible in **Suppl. Fig. 5**. We have also improved the presentation of the 3D model and placed it earlier in the manuscript (**Fig. 4**).*

A lot of neurons seem to be modulated at "nose down" or "nose up" (ND/NU) (Fig. 3). One can think that it is the simple consequence that in these positions the azimuth is undefined (or 'degenerate'). A pure "azimuth-tuned cell" would certainly fire for all azimuth directions in ND or NU position unlike in UP, when the cell is expected to fire for specific azimuth.

We understand that this a priori be a possible explanation, but it may be ruled out since a lot of 'tilt-only' cells are also modulated in ND/NU orientation, indicating that there is no direct link between the preference for ND/NU and azimuth tuning. We now mention this point in the discussion: "This over-representation is observed in tilt-only and well as in conjunctive cells, indicating that it is not linked to the 3D properties of azimuth tuning.". Note also that azimuth is degenerate in NU/ND only under the premise that cells encode azimuth in an EH frame – which is contradicted by our analyses, as well as by Page et al. Jeffery (2017) and Shinder and Taube (2019) data. In the TA frame, azimuth is degenerate in upside-down (UD) orientation, which could raise the same question since some neurons prefer upside-down. Yet, the same argument (that some tilt-only cells have the same preference) could be made.

It is not immediately clear how to reconcile these results with the hypothesis that the HD system is a 1d ring attractor. The present data seem to suggest otherwise, in particular that the system has additional possible states that are not fully explained by azimuthal tuning. The idea of the tilted azimuth (TA) is apparently that, indeed, the HD system is a 1d ring whose member neurons fire for angles relative to the orientation of the head in 3D. It is necessary to carefully explain the hypothesis here. The fact that pairwise differences of preferred direction in the azimuthal plane (Fig. S8) are preserved is a good indication that the activity may arise from a 1D ring attractor. However, along the same lines as the

previous comment, should the neurons be additionally modulated by tilt, pairwise correlation during sleep (As previously reported) could not be fully explained by their azimuth tuning. Otherwise, if the authors think that these observations are not incompatible, this point should be discussed.

We thank the reviewer for this very interesting comment. The fact that azimuth is encoded in TA frame doesn't challenge the concept of 1D ring attractor (since azimuth remains an 1D circular variable). However, the finding that most cells are conjunctive raises interesting questions. Indeed, if some cells prefer similar azimuth but different tilt positions, then they will necessarily have to fire at different points in time, contradicting the principle of a 1D attractor.

In the revised manuscript, we used data recorded on the platform (with a tilt angle of 60°) and performed a similar analysis as in Peyrache et al. 2015 (Suppl. Fig. 19). We found that pairwise correlations between azimuth-tuned cells (which are at the basis of the analysis in this study) do indeed decrease when the head tilts. We conclude that the HD network does form a 1D attractor when walking upright, but that its structure is more complex during 3D motion. Note that we didn't use data from the rotator in this analysis because responses are attenuated, which makes correlations too weak to conduct the analysis. Therefore, our analysis is limited to the range of tilts possible on the platform (60°). We would expect the correlation to decrease even further for larger tilt angles, and we certainly hope to explore this question in future studies.

The question remains of how to explain Peyrache's data when mice sleep. We think that the answer is likely that mice were sleeping near to upright position, such that responses recorded when awake and asleep were comparable.

The similarities with the bat HD system should be also carefully discussed since there are many differences. First, in bats, the recordings were performed in the post-subiculum. Furthermore, the tuning curves of the HD cells in the bat system were different from what is shown in the present manuscript. In bats, some cells fire for specific values of azimuth and pitch, for example. It was not the case here. Last, in bats, there is an anatomical gradient of tuning. Is there any sign that such anatomical distribution exists in mice?

We edited the discussion to carefully discuss these points (although they are addressed at different parts of the discussion). Regarding the difference between the tuning curves, we believe that the major difference is that azimuth responses are less attenuated in inverted orientation (which may suggest that bats 'solved' the issue of a singularity existing in the TA model and dual-axis rule; see Section: "Three-dimensional HDC across mammalian species").

Regarding your point "Furthermore, the tuning curves of the HD cells in the bat system were different from what is shown in the present manuscript. In bats, some cells fire for specific values of azimuth and pitch, for example. It was not the case here." We would like to point out that some of our cells fire for some specific values of azimuth and pitch (e.g. the one in Fig. 4d). However, in our study, the fact that azimuth responses attenuate when the head tilts means that such conjunctive responses are mainly observed in cells that prefer small tilt angles.

We now discuss possible relations between ADN and presubiculum (section: "Three-dimensional HDC across brain regions"). What is currently known from their connectivity is that presubicular

HDC are activated by the ADN and in turn project to the entorhinal cortex. However, some studies (e.g. Peyrache et al. 2017) hint that they carry additional sensory information. It would be exciting to discover that they have additional functions (e.g. integrating visual information, perhaps together with the RSC).

We didn't find any anatomical gradient of tuning in the ADN; however one should consider that the ADN is a nucleus, unlike the presubiculum. We didn't identify any statistically significant difference between granular and dysgranular RSC either. Finally, the CIN is a 'single point' for us, since we didn't record it at different locations along the antero-posterior axis.

Last but not least, the trough-to-peak duration of ADN spikes are very short, comparable to axonal spikes and/or fast-spiking neurons. Some ADN neurons are pyramidal thalamocortical neurons that should show trough-to-peak duration of at least 300ms. The examples in S1 are also a bit surprising. How come all waveforms are positive except on the highest-amplitude channel? Seems like it results from median or average filtering (i.e. the median/average of all channels is subtracted from each channel to remove noise). Would it be possible to show raw spike waveforms? Are spikes usually detected on more than one channel (axonal spikes are usually not too widespread)?

*All sorting was performed using an average filter to remove noise, mechanical artifacts and LFP. We recomputed the raw spike waveform for the example cell in **Suppl. Fig. 1**. Spikes in the CIN were generally detected on two channels (e.g. the blue cell in **Suppl. Fig. 1**).*

We typically observe a mixture of short trough-to-peak and long trough-to-peak neurons in the ADN. In a recent manuscript <https://doi.org/10.1101/684464>, we show that these neurons have similar properties when walking on an arena. In particular, HDC are equally distributed amongst both types of neurons, and the average firing and CV2 are similar in these groups; see Fig. 6 in this study. Furthermore, we re-analyzed data from Peyrache et al. 2015 and found a similar mixture of neurons with short trough-to-peak and long trough-to-peak, see Fig. 6 Suppl. 3 in the same manuscript (even though the bimodality is not as strongly pronounced in this data, possibly because they used probes instead of tetrodes and different filter settings). Therefore, although one may not be sure about the exact nature of the neurons emitting short-duration spikes, they are very similar to ADN neurons with long-duration spikes. We note that, to our knowledge, few studies report the duration of the spikes they record, and therefore neurons with short-duration spikes may be routinely included in ADN studies.

We realize that our sample included a majority of ADN neurons with short-duration spikes. We think this is a random occurrence: the 4 animals shown there indeed have a large percentage of short-duration spikes. However, 3D tuning is not restricted to this type of ADN neurons. In fact, in preliminary experiments, we recorded tilt-tuned and conjunctive cells in the ADN of 3 other animals. These recordings were performed in an initial version of the rotator where visual orienting cues were much weaker, and therefore we didn't include these animals in the main analyses of the preset study. Yet, since azimuth-tuned cells are identified using the arena and since tilt tuning is not affected by visual cues, we can classify these cells as azimuth-tuned, tilt-tuned, etc, in a manner equivalent to other cells in the study. Considering a broader data set, we find that 53%/47% neurons have short/long duration spikes respectively, and that the

proportions of conjunctive, tilt-only, azimuth-only and non-modulated cells are identical in both types of neurons.

Minor comments.

“In fact, 76/128 cells (14/22 ADN; 15/27 RSC; 47/79 CIN) were significantly ($p < 0.01$) more correlated to the gravity reference frame than the visual frame, and the remaining 52 cells were not significantly more correlated in either frame.” This is pretty unclear. What are these 52 cells correlated to then?

*Please see answers to Rev. 1. These are typically cells with weaker tuning. As a consequence, these cells are comparatively noisier, such that the correlation in both frames is lower and the correlation doesn't pass the significance threshold on a cell-by-cell basis (even though, at the population level, most of these cells are below the diagonal, i.e. correlate better with gravity). We show an example cell in **Suppl. Fig. 15n**.*

The authors use several tuning characterizations. It is unclear what the “Azimuth tuning amplitude” of Fig. 2 corresponds to: peak firing rate only or relative to baseline? What do the plots in Fig. 2 then show, expect that firing rates are overall preserved but slightly modulated by the conditions. Computing information per spike for each condition would have perhaps been more informative.

*The azimuth tuning amplitude is a peak-to-valley amplitude. However, we realize that it is not very informative since some cells have much higher firing rates than others. We have replaced it by a normalized tuning amplitude measure (peak-valley amplitude divided by peak) and added a clear definition of this measure in **Fig. 1**.*

*We considered computing a spatial information score (similar as in hippocampal place cells) but we found that we obtained very low values (ranging in 10^{-2} to 10^{-1} bits/s). It appears that, similar to the Rayleigh vector, this measure is highly influenced by the background firing rate. For instance, in the example tilt tuning curve in **Suppl. Fig. 7b**, the spatial information is only 0.17, but that it increases to 2.1 if the baseline firing rate is set to zero. Spatial information may therefore be better suited for quantifying cells such as hippocampal place cells. We are concerned that discussing spatial information may confuse the readers, and we didn't include it.*

****REVIEWERS' COMMENTS:**

Reviewer #1 (Remarks to the Author):

The authors addressed my comments and improved the clarity of the manuscript. I am therefore happy to recommend it for publication.

Reviewer #2 (Remarks to the Author):

I thank the authors for carefully addressing our comments. The manuscript now offers a very comprehensive analysis of the dataset and the data presentation has been improved. In my opinion, the manuscript is therefore suitable for publication.

Minor comments:

Fig S13, legend, last panel labels should be (f,g) and not (c,d)

Reviewer's comments

Reviewer #1 (Remarks to the Author):

The authors addressed my comments and improved the clarity of the manuscript. I am therefore happy to recommend it for publication.

Reviewer #2 (Remarks to the Author):

I thank the authors for carefully addressing our comments. The manuscript now offers a very comprehensive analysis of the dataset and the data presentation has been improved. In my opinion, the manuscript is therefore suitable for publication.

Minor comments:

Fig S13, legend, last panel labels should be (f,g) and not (c,d)

Author's reply

We thank the reviewers for their supportive comments. We corrected the legend of Fig S13.